# STEER-ME: Assessing the Microeconomic Reasoning of Large Language Models

## Abstract

Large language models (LLMs) are increasingly being applied to economic tasks like stock picking and financial analysis. Existing LLM benchmarks tend to focus on specific applications and often fail to describe a rich variety of economic tasks. Raman et al. (2024) offer a blueprint for comprehensively benchmarking strategic decision-making. However, their work failed to address the non-strategic settings prevalent in micro-economics. We address this gap by taxonomizing micro-economic reasoning into 58 distinct elements, each grounded in up to 10 distinct domains, 5 perspectives, and 3 types. The generation of benchmark data across this combinatorial space is powered by a novel LLM-assisted data generation protocol that we dub `auto-STEER`, which generates a set of questions by adapting handwritten templates to target new domains and perspectives. By generating fresh questions for each element, `auto-STEER` helps reduce the risk of data contamination, ensuring that LLM evaluations remain valuable over time. We leveraged our benchmark to evaluate 27 LLMs over each of the instantiated elements, examined their ability to reason through and solve microeconomic problems and compared LLM performance across a suite of adaptations and metrics. Our work provides insights into the current capabilities and limitations of LLMs in non-strategic economic decision-making and a tool for fine-tuning these models to improve performance.

## 1 Introduction

There is much recent interest in using language models (LLMs) to reason about economic topics. Some prominent examples include financial sentiment analysis, where LLMs are tasked with analyzing the sentiment information of financial texts (Malo et al., 2013; Maia et al., 2018; Araci, 2019; Yang et al., 2020); Named Entity Recognition, which asks the model to detect critical financial entities such as persons, organizations, and locations (Salinas Alvarado et al., 2015; Shah et al., 2022); financial text summarization, which entails condensing long unstructured financial texts into short summaries that capture crucial information and maintain factual consistency with the original long texts (Mukherjee et al., 2022; Zhou et al., 2021); and question answering, where LLMs are tasked with answering an economic question based on the provided information (Maia et al., 2018; Chen et al., 2021; 2022; Shah et al., 2022; Xie et al., 2023b; Raman et al., 2024). More open-ended applications are also starting to emerge. LLMs such as WallStreetBERT, TradingGPT, FinGPT, FinTral, and BloombergGPT are already giving advice to investors and financial advisors (Xie et al., 2023a; Li et al., 2023; Yang et al., 2023; Bhatia et al., 2024; Wu et al., 2023a). LLMs can help to automate budgetary planning and allocation (Chen et al., 2023). LLMs are also being deployed as agents in simulations to analyze the impact of policy changes on key indicators like inflation and GDP growth (Carriero et al., 2024; Li et al., 2024a).

Before LLMs should be trusted in such open-ended applications, they should demonstrate robustly strong performance on the fundamentals of economic reasoning (just as, e.g., financial advisors, budget planners, and economists are required to do). Many existing benchmarks have been proposed, many of which were introduced in papers cited above. However, most of these are quite narrowly focused on a single task and/or application, rather than assessing economic reasoning more broadly. A second—useful but insufficient—category of benchmarks tests foundational concepts in mathematics, ranging from basic arithmetic to complex problem-solving tasks (Huang et al., 2016; Ling et al., 2017; Amini et al., 2019; Lample & Charton, 2019; Zhao et al., 2020). Notable benchmarks include

GSM8K (Cobbe et al., 2021), a small but varied dataset that contains moderately difficult math problems and MATH (Hendrycks et al., 2021c), a challenging benchmark for which no evaluated model has yet attained expert-level performance across any of the 57 tested scenarios.

What might it look like to assess an LLM's economic reasoning more comprehensively? Economics encompasses a wide array of problems, such as determining optimal consumption bundles, forecasting profit in the face of uncertainty, or analyzing how a shift in supply impacts equilibrium prices and quantities. Each of these problems can occur in a wide range of contexts such as labor markets, consumer product markets, financial markets, or public policy. Beyond the breadth of inputs that must be considered, evaluating LLMs presents further challenges to benchmark designers. There is no guarantee that an LLM will perform equally well on problems that appear similar or are conceptually related (e.g., Hendrycks et al., 2021a). For instance, an LLM that excels at maximizing profit may struggle with minimizing cost. Similarly, LLMs can be susceptible to perturbations in the text of a question, which can impact their performance on otherwise similar problems (Ribeiro et al., 2020). For example, LLMs may excel in allocating budgets as a doctor, but struggle to allocate budgets as an educator. Finally, LLMs may reason correctly about their own incentives, but fail to apply this logic to other participants and hence have difficulty understanding market or aggregate level responses (e.g., total supply, demand, and prices). Therefore, in order to be comprehensive, a micro-economic benchmark must exhibit broad variation across problems, contexts, and textual perturbations. It is similarly nontrivial actually to conduct experiments that comprehensively assesses how well different LLMs perform at economic reasoning tasks. Different models may leverage distinct architectures, driving performance differences (Sanh et al., 2020; Islam et al., 2023; Raman et al., 2024). Additionally, adaptation strategies—such as fine-tuning, prompt engineering, and output distribution modification—can dramatically influence a model's effectiveness (Brown et al., 2020; Lester et al., 2021; Kojima et al., 2023). Under the right adaptations, models with as few as 7B parameters can achieve state-of-the-art performance (e.g., Bhatia et al., 2024). Furthermore, robustness across multiple task formats (e.g., multiple-choice QA, free-text QA, etc.) is crucial for understanding the gaps in an LLM's reasoning capabilities. A model that performs well on one task format may underperform on others, which suggests gaps in its reasoning processes. Finally, scoring performance using only a single metric can give a skewed understanding of an LLM's abilities and limitations (Schaeffer et al., 2023), or obscure tradeoffs that are relevant to practitioners (Ethayarajh & Jurafsky, 2020). Without a comprehensive evaluation, we risk misattributing performance to a LLM when it is instead driven by an adaptation strategy or is an artifact of the metric used.

A recent paper by Raman et al. (2024) developed a benchmark distribution for assessing economic reasoning in strategic settings that aims for comprehensiveness in the senses just described. This work serves as a starting point for our own paper, and so we describe it in detail. First, they developed a taxonomy that divided the space of game theory and foundational decision theory into 64 distinct "elements of economic rationality," ensuring that the elements in the benchmark covered a wide range of strategic contexts and decision-making problems. Second, they formalized a hierarchy across elements so that an LLM's performance could be better understood in the context of its dependent subtasks. They generated a huge set of questions from this taxonomy, dubbed STEER, which vary in their difficulty and domain (e.g., finance, medicine, public policy). Finally, they evaluated a spectrum of LLMs over two adaptation strategies and scored with a suite of metrics. They defined this evaluation framework as a STEER Report Card (SRC), a flexible scoring rubric that can be tuned by the user for their particular needs.

A key drawback of STEER is that, in its focus on game-theoretic reasoning, it neglects much of the subject matter of microeconomics: multiagent settings in which agents nevertheless act nonstrategically. Such reasoning is widespread in competitive markets, where each agent's impact on the market is too small to affect prices unilaterally. For example, while a mobile phone manufacturer might make a strategic decision about the number of handsets to produce and the price to sell them at, a small farm's decision to produce wheat instead of corn given market prices is non-strategic. We employ—and expand upon—the STEER blueprint to construct a benchmark for testing LLMs on economics in non-strategic environments. Following Raman et al. (2024), we first identified a taxonomy of 58 elements for non-strategic economics. We then instantiated each element in the taxonomy across 8–10 domains and up to 2 types. From here, we expanded on the blueprint in two ways. First, we increased the diversity of the questions in the dataset and instantiated each element in 5 different *perspectives* and up to 3 *types* (as defined in Section 3.1). Second, we expanded their evaluation framework to include newer LLMs (27 in total), some new adaptations (3 that we

developed and 2 more from the literature) adaptations, and many new scoring metrics (a family of 4 calibration metrics). We dub our benchmark STEER-ME.

Even given the best possible LLM benchmark, data contamination poses an increasingly important challenge (Sainz et al., 2023; Deng et al., 2023; Ravaut et al., 2024). Data contamination occurs when the test data used to evaluate an LLM is similar or identical to data the LLM encountered during training, leading to inflated performance metrics that do not accurately reflect the LLM's true capabilities. To tackle this issue, we introduce a new dynamic data generation process called auto-STEER which we used to generate all of the questions in STEER-ME. auto-STEER combines many of the features present in existing dynamic and modular frameworks (Gioacchini et al., 2024; Wang et al., 2024; White et al., 2024) that we detail in Appendix B.

In what follows, Section 2 gives an overview of our taxonomy; for space reasons we defer definitions and examples of each element to Appendix A. Section 3 describes how we used this taxonomy to build the benchmark distribution. For 37 elements, we have written LLM prompts to synthetically generate $1,000$–$5,000$ multiple-choice questions and manually validated 500 generations per element. Section 4 describes the setup of an experiment in which we generated full SRCs for 27 LLMs, ranging from Llama-2 7B to GPT-4o, evaluated on a total of $21,000$ test questions. We spent $\$5,896.33$ making requests to OpenAI and Anthropic's API and $6.81$ GPU years of compute to evaluate open-source models.

Finally, we discuss the results in Section 5. Here, we offer a few highlights. We observed a significant variation in performance across both LLMs and elements. Even among large models, most underperform on at least a few tasks, indicating that size alone is not a sufficient predictor of success across our benchmark. The one exception is o1-preview, which consistently achieved top performance on every element we tested, standing out as the most robust and accurate model in our evaluations. Across domains and perspectives, LLMs generally exhibited stable performance, although certain elements, particularly those testing conceptual understanding of economic principles, exposed weaknesses in even the more advanced LLMs. Additionally, we observed considerable variation in LLM performance across different adaptation strategies. For instance, when models were not able to view the options prior to answering, performance dropped significantly. This performance gap further underscores a general reliance on external cues and hints at limitations in the ability to independently derive solutions from first principles.

We release all model outputs to support evaluation research and contributions, and provide a public website with all results, underlying model predictions details, alongside an extensible codebase to support the community in taking STEER-ME further.

## 2 ELEMENTS OF ECONOMIC RATIONALITY

Our first step in generating a benchmark for non-strategic microeconomics is to taxonomize this space. Previous work by Raman et al. (2024) developed a taxonomy for economic rationality within strategic domains. Their approach involved identifying foundational principles that define how agents should make decisions in specific environments and then organizing these principles, or "elements," into progressively more complex decision-making scenarios. We adopt a similar hierarchical approach for STEER-ME, focusing on organizing economic decision-making principles into structured categories. However, unlike STEER, which assesses decision-making in strategic environments, our focus is assessing how agents make decisions given prices and quantities that are determined by the forces of supply and demand. We call this sub-field non-strategic microeconomics.

Two of the settings from STEER remain directly relevant to non-strategic microeconomics: FOUNDA-TIONS and DECISIONS IN SINGLE-AGENT ENVIRONMENTS. As we describe our taxonomy, we begin with these foundational settings. The elements we incorporate from FOUNDATIONS—arithmetic, optimization, probability, and logic—are core mathematical skills essential for microeconomic reasoning and are already present in STEER. In STEER-ME, we expand this setting by adding elements that test basic calculus, such as single-variable derivatives and linear systems of equations. In STEER, DECISIONS IN SINGLE-AGENT ENVIRONMENTS focused on testing whether an agent can adhere to the von Neumann-Morgenstern utility axioms when making decisions over a set of alternative choices. We include those axiomatic elements and extend this setting to include testing the properties

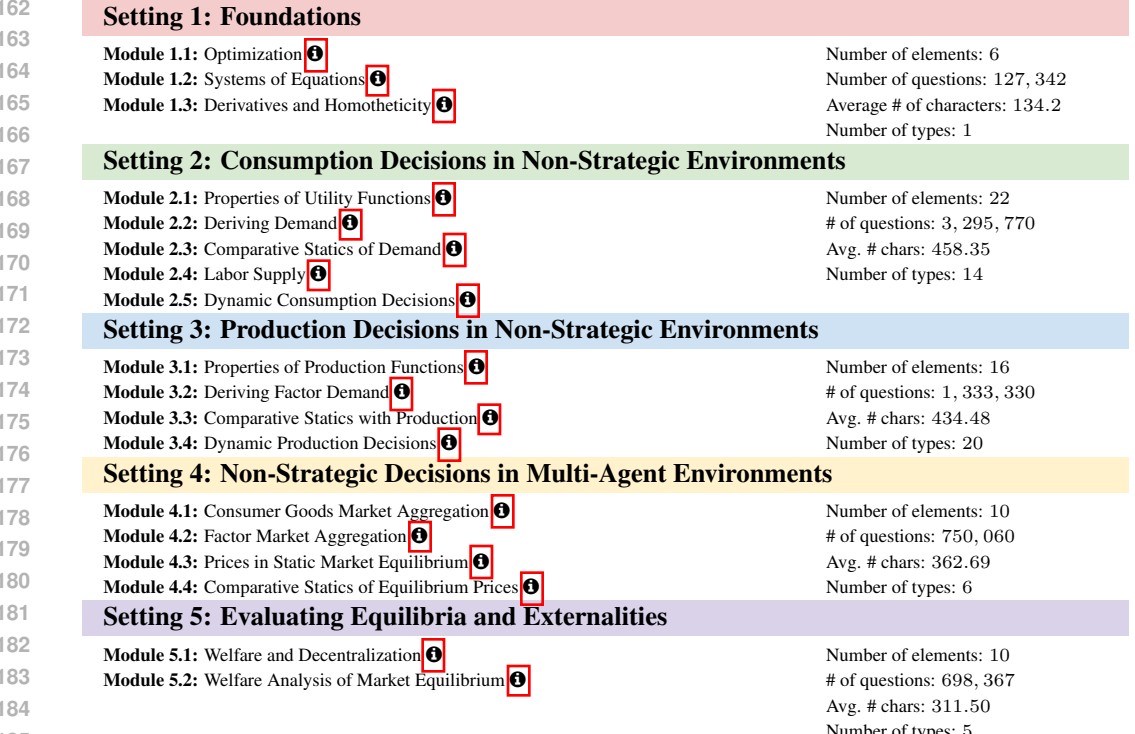

**Setting 1: Foundations**

**Module 1.1:** Optimization
**Module 1.2:** Systems of Equations
**Module 1.3:** Derivatives and Homotheticity

Number of elements: 6
Number of questions: 127, 342
Average # of characters: 134.2
Number of types: 1

**Setting 2: Consumption Decisions in Non-Strategic Environments**

**Module 2.1:** Properties of Utility Functions
**Module 2.2:** Deriving Demand
**Module 2.3:** Comparative Statics of Demand
**Module 2.4:** Labor Supply
**Module 2.5:** Dynamic Consumption Decisions

Number of elements: 22
# of questions: 3, 295, 770
Avg. # chars: 458.35
Number of types: 14

**Setting 3: Production Decisions in Non-Strategic Environments**

**Module 3.1:** Properties of Production Functions
**Module 3.2:** Deriving Factor Demand
**Module 3.3:** Comparative Statics with Production
**Module 3.4:** Dynamic Production Decisions

Number of elements: 16
# of questions: 1, 333, 330
Avg. # chars: 434.48
Number of types: 20

**Setting 4: Non-Strategic Decisions in Multi-Agent Environments**

**Module 4.1:** Consumer Goods Market Aggregation
**Module 4.2:** Factor Market Aggregation
**Module 4.3:** Prices in Static Market Equilibrium
**Module 4.4:** Comparative Statics of Equilibrium Prices

Number of elements: 10
# of questions: 750, 060
Avg. # chars: 362.69
Number of types: 6

**Setting 5: Evaluating Equilibria and Externalities**

**Module 5.1:** Welfare and Decentralization
**Module 5.2:** Welfare Analysis of Market Equilibrium

Number of elements: 10
# of questions: 698, 367
Avg. # chars: 311.50
Number of types: 5

Table 1: High-level diagram of the taxonomy of elements of rationality. At the top level, we divide the space of decision making into 5 settings; we further subdivide settings into modules (e.g., Comparative Statics of Demand) that capture conceptually similar behaviors. We also include a few summary statistics about the dataset.

of commonly used parameterizations of utility functions in non-strategic microeconomic contexts, such as utility functions with satiation points, monotone preferences, and budget constraints.

Building directly on these foundational settings, we introduce the next setting, DECISIONS ON CONSUMPTION IN NON-STRATEGIC ENVIRONMENTS, which tests an agent's ability to optimally exchange time and money for desired goods and services. Elements in this setting assume that the agent is a price taker, meaning that the agent accepts market prices as given rather than forecasting how a purchase might move the market. First, we test the agent's ability to derive demand functions consistent with the axioms and functional forms from DECISIONS IN SINGLE-AGENT ENVIRONMENTS. These foundational elements are useful in assessing whether an agent can make consistent, rational choices in response to market prices. We then include elements testing the agent's ability to determine optimal consumption bundles, decide when to leave the workforce, and conduct comparative statics with demand functions.

DECISIONS ON PRODUCTION IN NON-STRATEGIC ENVIRONMENTS tests an agent's ability to decide on the combination of inputs to efficiently produce goods and services to maximize their profits. The setting starts by assessing the agent's ability to identify and analyze basic properties of production functions, such as the relationship between input quantities and output levels. This includes concepts like returns to scale, diminishing marginal returns, and the technological constraints that shape production capabilities. We then test the agent's ability to conduct expenditure minimization and its dual, profit maximization. This involves solving optimization problems where the agent must use marginal analysis to determine the quantity of output that maximizes profit (i.e., minimizes cost).

DECISIONS IN MULTI-AGENT NON-STRATEGIC ENVIRONMENTS considers consumers and producers who each reason according to the principles just described to trade with each other. This more complex setting requires an agent to reason about how the aggregated behaviors of consumers and producers lead to market-clearing prices that balance supply and demand. This setting covers

elements such as finding market-clearing prices, computing competitive equilibria, and analyzing the comparative statics of equilibrium in markets where individual actions do not directly impact others.

Our last setting, EVALUATING EQUILIBRIA AND EXTERNALITIES, tests agents on their ability to evaluate whether equilibria are efficient and to analyze the effects of interventions, such as taxes or price ceilings, on welfare. In this setting, agents must not only be able to analyze how supply and demand dynamics establish equilibrium prices but also consider how external interventions shift these dynamics and alter the behavior of both consumers and producers. The elements in this setting can be relatively simple (e.g., compute consumer/producer surplus) or involve detailed counterfactual analysis (e.g., predict how interventions impact prices, the allocation of resources, and welfare outcomes).

For a more detailed discussion on the structure of these elements and the methodology we used to group the elements, including formal definitions, we refer the reader to Appendix A.

## 3  THE STEER-ME BENCHMARK

We first give an overview of STEER-ME dataset and then explain the process we used to generate and validate these questions, which we call auto-STEER. Finally, we describe our evaluation framework.

### 3.1  DATASET

We adopted the widely used Multiple-Choice Question Answering (MCQA) format for our benchmark (see, e.g., Rajpurkar, 2016; Wang et al., 2018; 2019; Zellers et al., 2019; Hendrycks et al., 2021b; Shah et al., 2022; Liang et al., 2022; Suzgun et al., 2022). In this format, each test question presents a decision-making scenario along with several candidate options, where only one is correct. As an evaluation paradigm, a benefit of MCQA is that it provides a standardized way to evaluate an LLM's ability to correctly respond to given prompts. MCQA tasks have well-established metrics like exact-match accuracy or expected calibrated error that provide interpretable measures of how well an LLM answers questions (Liang et al., 2022; Li et al., 2024b). Furthermore, many real-world applications of LLMs in economics involve answering questions: e.g., chatbots (Inserte et al., 2024) and virtual assistants (BloombergGPT Wu et al., 2023b).

Our own benchmark consists of a total of 30 instantiated elements, each containing 5000-20, 000 MCQA questions. Each question is characterized by a (type, domain, perspective) tuple. Different *types* represent distinct ways of testing an agent's abilities within an element. For example, we could assess an agent's ability to perform profit maximization by asking "What is the maximum profit?" or "How much labor is needed to maximize profit?" The *domain* of a question indicates which of 10 predefined topic areas it pertains to: consumer goods, medical, finance, education, technology, entertainment, environmental policy, politics, sports, or gambling. Finally, the *perspective* of a question represents which of the 5 predefined perspective the question was written in: first-person, second-person, third-person anonymous, third-person female and third-person male. We skip over (type, domain, perspective) combinations that do not lead to coherent questions; for example, questions about welfare theorems do not make sense in gambling settings.

### 3.2  AUTO-STEER

Like Raman et al. (2024), we leveraged a state-of-the-art LLM to help generate our dataset. We substantially extended their methodology, however, by adding an additional style-transfer step where we asked the LLM to rewrite questions in new domains or perspectives. This greatly increased the variety of questions we were able to add. This section describes how we used our new approach to design STEER-ME.

First, for each type we hand-wrote a set of gold-standard example templates that served as the seeds for the data generating process. As can be seen in Figure 12, these templates were tagged with a domain, a perspective, and a type, if appropriate. The majority of these questions had *labeled fields* for numbers (e.g., "...the cost of labor is {cost}...") which were programmatically filled for test time. See Figure 1 for an example.

Next, we asked the LLM to style-transfer these templates into each of the domains. Our prompt included explicit instructions to maintain the same set of labeled fields as the hand-written templates. Figure 13 depicts the style-transfer page in our web application along with the prompting instructions. LLMs can be inconsistent in maintaining the economic meaning of questions after domain style transfer, so we hand-checked each of the outputted templates and edited them when necessary. This was all done in the web application: see Figure 15. We then further style-transferred each of these newly generated templates into each perspective, resulting in up to 40 unique domain-perspective pairs for each type. We ran an additional check on the style-transfer process by filling the labeled fields in the templates with values and asking the LLM to solve the questions as written, which we found could highlight mistakes in question wording or in programmatically filled values. See Figure 14. (We were careful only to use his procedure to correct mistakes in the templates, not to tune the difficulty of the questions in a way that would bias our benchmark.)

We then took each of these templates and asked the LLM to replicate the template, keeping the domain, perspective and labeled fields fixed but modifying exact words or objects used in the question. We generated 100 new templates for each element, crossing every domain and perspective pair, resulting in 30,000 templates across the dataset. We then spot-checked 500 of the resulting templates for each element, and flagged 99.88% of the templates as valid.

Finally, we created 20 instantiated questions from each template by filling its labeled fields with randomly generated values. We restricted the random generator to output numbers that were appropriate given the context: e.g., demand functions had negative slopes, positive values for equilibrium prices, etc. We programmatically solved each question and filled in the appropriate options and answer. In the end, we produced 1,000 questions per (domain, perspective) pair and up to 40,000 per type.

**Question:**

Sophie is buying textbooks for her university classes, her demand for textbooks at any given price is expressed by the following demand function {d_function}. What is Sophie's consumer surplus if the price of textbooks is {price}?

Domain: Education, Perspective: Third Person Woman

**Question:**

John is purchasing hockey sticks, his demand for hockey sticks at any given price is expressed by the following demand function {d_function}. What is John's consumer surplus if the price of hockey sticks is {price}?

Domain: Sports, Perspective: Third Person Man

Figure 1: This figure depicts two questions in the consumer surplus element with different domains and perspectives. The text colored in red are the labeled fields that will be filled for test time and the text in blue is the perspective. On top, a question is framed in the education domain from a third-person woman perspective, while on the bottom, the same question is written for the sports domain from a third person man perspective. These were both generated during the style-transfer step in the data generation process.

### 3.3 EVALUATION FRAMEWORK

We now turn to describing our evaluation framework. Following other work in this space, we consider an LLM as a black box to which we provide inputs in the form of prompts (i.e., strings) and adjust the decoding parameters (e.g., temperature) to analyze the resulting output completions (i.e., strings) and log probabilities, when available. Within this black-box framework, we consider two classes of adaptations: performance adaptations, which modify inputs to affect performance on a task, and diagnostic adaptations, which aim to analyze specific behaviors or model characteristics. We then score LLMs across a suite of metrics.

We follow Raman et al. (2024) by allowing a user to tune the evaluation framework for their specific needs by choosing for their set of LLMs: the set of elements in the evaluation, the adaptation chosen for each LLM and a scoring metric. For instance, one may only want to evaluate specific economic modules in our taxonomy (e.g., utility maximization for individual decision-making in DECISIONS ON CONSUMPTION IN NON-STRATEGIC ENVIRONMENTS or production optimization scenarios in DECISIONS ON PRODUCTION IN NON-STRATEGIC ENVIRONMENTS), or conduct comparative assessments across adaptation strategies, or evaluate targeted use cases like medical or financial decision-making. We provide a number of predefined evaluation frameworks in our web application as well as allowing users to create new evaluation frameworks.

We classify any adaptation as a performance adaptation when the inputs are modified in a way that is intended to increase an LLM's performance on a task. Common performance adaptations are chain-of-thought reasoning (Wei et al., 2022; Yoran et al., 2023; Huang et al., 2023; Kojima et al.,

2023) and few-shot prompting (Brown et al., 2020; Perez et al., 2021). We focus on zero-shot chain-of-thought reasoning.

**Zero-Shot Chain-of-Thought (0-CoT).** There has been work showing that performance can be improved by asking an LLM to explain its reasoning before outputting an answer (Wei et al., 2022; Yoran et al., 2023; Huang et al., 2023; Kojima et al., 2023). We follow Kojima et al. (2023) in implementing 0-CoT by first asking the LLM to explain its reasoning and then subsequently asking it to select the correct answer. We take two approaches to adapting 0-CoT to MCQA, which we denote *hidden* and *shown*. In the hidden approach, we give the LLM the question text and ask it to explain its reasoning—we only provide the candidate options in the second step. In the shown approach, the LLM is given both the question text and candidate options when it is asked to explain its reasoning. See Figure 11 in the appendix for an example.

### 3.3.1 DIAGNOSTIC ADAPTATIONS

*Diagnostic adaptations* alter the prompt or decoding parameters not to improve performance, but rather to gain a better understanding of an LLM's behavior.

**Calibrated Answer Replacement (CAR).** In CAR, we modify the candidate options by replacing one of the options with the following string: "No other option is correct." For a test containing questions with $n$ options, we replace the correct answer with this placeholder in a $1/n$ fraction of questions. For the remaining questions, we replace one of the incorrect answers instead. This ensures that an LLM that always chooses "No other option is correct" receives the same accuracy as random guessing.

**Reshaped Probability Mapping (RPM).** Sometimes, LLMs can assign nonzero probability to tokens that do not correspond to any of the options available. Such errors are trivial to fix in any downstream application. However, if not corrected for, such errors can distort performance metrics, e.g., leading models to appear to perform worse than random guessing. We call the adaptation that addresses this issue RPM and take two approaches to reshaping the outputs. The first approach is conditioning the output distribution to only valid options. However, in cases where the model puts very little weight on *any* correct option this renormalization can make the model appear overconfident. Our second approach attempts to deal with this by mixing the output distribution with a uniform distribution over valid options, this means if very little probabilistic mass is given to any correct option its output will look more uniform and hence less confident in its answer. We define these adaptations and offer further discussion in Appendix C. Importantly, neither implementation changes which of the valid option tokens receives the largest weight in the output distribution, and therefore the LLM's accuracy.

**Free-Text QA.** In addition to the diagnostic adaptations discussed earlier, we conducted experiments involving free-text generation question answering to more closely align with real-world use cases. We ask an evaluator LLM to report the answer the chain of thought reasoning arrived at and None if there is no easily findable answer. We then scored a model's answer as correct if it was within 98% of the correct answer value and is closer to the correct answer than any other option. We include the prompt we used in Appendix C.3.

### 3.3.2 SCORING

Given a complete set of model responses, it is far from straightforward to choose a way of computing a single, overall performance score. Consequently, benchmarks often employ a suite of metrics to provide a more comprehensive assessment of performance (Wang et al., 2019; Gehrmann et al., 2021; Liang et al., 2022; Srivastava et al., 2023). We evaluate LLMs using three categories of metrics: accuracy, calibration, and robustness. We leave the discussion and definitions of our scoring metrics in the Appendix D and simply list the metrics below:

- Accuracy: Exact-match accuracy and Normalized accuracy
- Calibration: Expected calibration error, Brier Score, and Expected Probability Assignment
- Robustness: Domain Robustness and Type Robustness

We score LLMs on their restricted output distribution over valid option tokens, modified using the diagnostic adaptation RPM as described in Section 3.3.1. For each model, we also report the proportion of responses where the top token is not a valid option token.

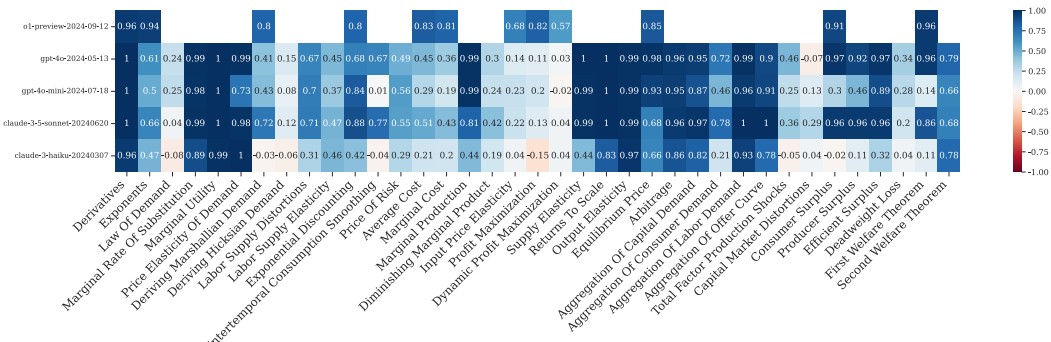

Figure 2: This figure plots a heatmap of the closed-source LLM performance measured with normalized accuracy on the 30 elements we instantiated. The LLMs, on the y-axis, are sorted in terms of parameter size. The elements, on the x-axis, are grouped by setting.

A LLM's score on an element is the average taken over all questions in an element. We consider an element a base concept in our benchmark and therefore define the accuracy and confidence metrics with respect to an element.

## 4 EXPERIMENTAL SETUP

Table 7 in the appendix lists the 15 LLMs we evaluated. We ran gpt-4o, gpt-4o-mini, and o1-preview using OpenAI's API (OpenAI, 2020); claude-3-5-sonnet and claude-3-haiku using Anthropic's API (Anthropic). We obtained 10 open-source LLMs from the HuggingFace Hub (Wolf et al., 2019) and ran them on between 1 and 4 A100, Tesla M60, and V100 GPUs (depending on model size) on one of several dedicated compute clusters to which we have access.

In multiple-choice classification, there are a few ways one might represent the input to an LLM. We follow prior work by Hendrycks et al. (2020) who introduced the *joint* approach where all answer choices are combined with the question into a single prompt, and the LLM predicts the most likely option letter[1]. We then decoded valid multiple choice responses from all LLMs as described in Section 3.3.2. For those LLMs where we had no access to the output distribution (claude-3-5-sonnet, claude-3-haiku, o1-preview) we took the top token.[2] In the free-text QA adaptation, we used gpt-4o-mini as the evaluator LLM due to its low cost and high performance in text retrieval.

Due to time and budget constraints we evaluated the closed-source LLMs, claude-3-5-sonnet, claude-3-haiku, gpt-4o, and gpt-4o-mini, on all 30 of the instantiated elements, all open-source models on 20 of the instantiated elements, and o1-preview on 13 elements. We applied our benchmark across all combinations of adaptations and LLMs, except for in the case of o1-preview. We did not explicitly ask o1-preview to conduct 0-CoT reasoning because it is a reasoning model and simply asked for the top token. Consequently, we did not run o1-preview on the hidden implementation of 0-CoT. This led to a total of 4 experiments for o1-preview and 8 for all other LLMs.

## 5 RESULTS

Figure 2 depicts aggregate performance across our whole benchmark, using normalized accuracy with the shown implementation of 0-CoT and without CAR. We chose these adaptations as we observed that LLMs performed the best on that adaptation configuration on average. We plot the models in descending order of parameter size and the elements in taxonomical order (i.e., FOUNDATIONS elements first) breaking ties alphabetically. Due to space constraints we only include LLMs that performed sufficiently better than random guessing: with normalized accuracy greater than 0.2 on

---

[1]There is another approach, called *separate* and employed by Brown et al. (2020). However, this approach is better suited to tasks where the answer choices are long-form generations.

[2]OpenAI models only return the top 20 tokens, however, we never saw a valid option token not present in those top 20 tokens.

average (see Figure 3 in the appendix for the remaining models). Furthermore, we observed that for the LLMs that we plot, our calibration metrics were correlated with normalized accuracy and hereafter focus mainly on normalized accuracy.

Elements across the settings in our benchmark proved to be difficult from FOUNDATIONS to EVALUATING EQUILIBRIA AND EXTERNALITIES, however, on the 13 elements that were tested, o1-preview was the most accurate model (see the top row in Figure 2). Even in elements where every other model was close to random guessing (e.g., Profit Maximization and Dynamic Profit Maximization) o1-preview obtained high accuracy. Besides o1-preview, no LLM consistently outperformed other LLMs across our benchmark.

A common struggle for LLMs was the precision required to solve optimization problems, particularly those that involve multiple sequential steps of computation and economic interpretation. For instance, in a challenging task like Dynamic Profit Maximization, LLMs are tasked with solving a 2-stage optimization problem that requires accurately performing a series of interdependent calculations. Each step, from identifying the correct approach to interpreting the economic implications and executing precise computations, presents opportunities for errors to accumulate.

However, even elements with simple mathematical problems presented opportunities for errors. None of the closed-source LLMs, except for o1-preview were able to consistently compute the Deadweight Loss of a Monopoly; an element whose primary mathematical requirement is computing the area of a triangle. We discovered that models like claude-3-5-sonnet and gpt-4o often used an incorrect formula for computing deadweight loss and made errors in interpreting the marginal cost, a crucial step in the problem-solving process. To better understand these errors we investigated model performance in the free-text QA adaptation. Figure 4 and Figure 5 show the distribution of correct responses and specific errors for claude-3-5-sonnet and gpt-4o, respectively. While gpt-4o displayed performance better than random guessing, errors stemming from the use of an incorrect formula consisted of the majority of responses. claude-3-5-sonnet, on the other hand, exhibited a higher prevalence of incorrect formula errors, with nearly 44% of its responses relying on a particular incorrect formula for deadweight loss. Furthermore, gpt-4o was more susceptible to compounding issues, incorrectly computing marginal cost and using an incorrect deadweight loss formula, than claude-3-5-sonnet. We describe these errors in more detail in Appendix I.2.

## 5.1 ROBUSTNESS

*Domain Robustness.* While overall the variation across domains was limited, we observed noticeable differences in specific elements. In particular, elements testing conceptual understanding of foundational principles (e.g., first welfare theorem) showed that certain domains provided more effective contextual cues for the LLMs. For example, in the consumer goods domain—where items like apples, chairs, or mugs are familiar in economic word problems—LLMs were more likely to recognize the task as an economic problem and anchor their reasoning in classical economic principles.

In contrast, the technology domain, where the economic context could be interpreted as a real-world scenario presented more challenges. The LLMs often failed to recognize what was being asked and equivocated when reasoning about the problem. The largest performance gaps appeared in the First Welfare Theorem and Second Welfare Theorem elements. For instance, claude-3-5-sonnet exhibited a gap of $0.657$ in accuracy between the consumer goods and technology domains, claude-3-haiku had a gap of $0.48$, and gpt-4o-mini showed a gap of $0.278$.

*Type Robustness.* Here, we examine LLM performance across different families of functions used in economic reasoning. These include Cobb-Douglas, Leontief, linear, and non-linear functions. Each family of functions poses distinct challenges depending on the mathematical operations and economic concepts being tested. While Cobb-Douglas functions are ubiquitous in economics, they can often be more challenging for language models as they feature non-integral exponents, which add a layer of difficulty in operations like differentiation. For instance, in Figure 6, we observe that, with the exception of claude-3-haiku, performance on non-linear functions (polynomials with integer exponents of degree $\leq 3$) surpasses performance on Cobb-Douglas functions.

For any given element, the family of functions that is the most difficult can vary. For example, computing the Returns to Scale of a Cobb-Douglas production function is the sum of the exponents and computing the Output Elasticity corresponds to the exponent on the input.

## 5.2 Adaptations

We observed that in the hidden implementation, LLM performance was worse overall compared to the shown implementation. This suggests that LLMs benefit from being able to reason directly over the options.

One pattern we observed was models exploiting the provided options to "cheat" the question. Instead of deriving the answer from first principles, LLMs would insert the candidate options directly into functions in the question text and select the correct answer based on which option produced the best result. This strategy was particularly prevalent in the Profit Maximization element, where models were asked to find the amount of labor to employ that maximizes a profit function. While the intended approach was for the model to take the derivative of the profit function and identify the profit-maximizing labor, LLMs often bypassed this by simply plugging in each of the given options and selecting the one that resulted in the highest profit. We observed this behavior in every question that we spot-checked where gpt-4o answered correctly (see Appendix E.3).

We also found that the inclusion of options could signal how to reason about the question. This was particularly prevalent in the aggregation elements in EVALUATING EQUILIBRIA AND EXTERNALITIES and especially in the Aggregation of Consumer Demand element, which ask models to aggregate the quantity demanded for some number of consumers. In the hidden implementation, models often failed to multiply the quantity demanded by the number of consumers in the market. When presented with the options, the additional signal in the magnitude of each of the candidate options increased performance. Providing evidence of this, we found that as the number of digits in the answer increased so too did the exact-match accuracy. Figure 7 (in the appendix) shows that as the number of digits in the answer increased, so too did the exact-match accuracy, providing evidence that models use the magnitude as a hint for reasoning. We show an example of this behavior in Appendix E.1.

To further investigate this effect, we examined four elements (Intertemporal Consumption Smoothing, Profit Maximization, Aggregation of Consumer Demand, and Producer Surplus) that exhibited the largest gap in accuracy between hidden and shown adaptations. Our analysis revealed that performance was almost always worse under the free-text QA adaptation compared to the hidden adaptation, see Figure 9. This performance gap appears to stem from the models' tendency to selecting the closest option to the free-text answer. Figure 10 shows the percentage of times that models were correct under the hidden adaptation but incorrect under the free-text adaptation due to guessing the closest answer. In almost all cases the majority of the gap is due to this phenomenon. We offer more discussion in Appendix I.3.

## 6 Discussion and Conclusions

Our work introduces a novel benchmark specifically designed to evaluate LLMs' performance in non-strategic microeconomics, focusing on tasks that require a deep understanding of optimization, marginal analysis, and economic reasoning in individual decision-making contexts. This benchmark provides a comprehensive tool to assess the strengths and weaknesses of current models, revealing where they excel and where they struggle in applying foundational economic concepts. By identifying these areas, our benchmark can guide users in determining when LLMs can be trusted to perform well in economic analyses and when further development is needed.

In cases where models fall short, our benchmark serves as a practical resource for targeted improvements—whether through fine-tuning models, curating more specific datasets, or developing architectures better suited for microeconomic reasoning. These enhancements have the potential to impact a variety of economic applications, such as simulating consumer behavior, analyzing market dynamics, or conducting policy evaluations.

Looking ahead, we plan to expand our benchmark by incorporating additional elements from the microeconomics literature, deepening the evaluation of non-strategic decision-making. We encourage suggestions on new elements to include and make `auto-STEER` public for others to add more elements or expand on the elements we have currently. We also intend to explore further experimentation with additional LLMs, adaptation strategies, and prompt configurations, along with more detailed analyses of model performance.

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
