## A TAXONOMY OF NON-STRATEGIC MICROECONOMICS

### A.1 DECISIONS ON CONSUMPTION IN NON-STRATEGIC ENVIRONMENTS

We begin by characterizing the space of elements that test an agent's ability to optimally allocate their limited resources to goods and services they desire. In economics and decision theory, the most primitive approach to describing the preferences of decision-makers is to use a function that maps a set of possible choices to the agent's optimal choice within that set. Under a set of intuitive assumptions, such as *transitivity* (i.e., if bundle $X$ is preferred to bundle $Y$, and $Y$ is preferred to bundle $Z$, then $X$ must be preferred to $Z$), it becomes possible to "rationalize" preferences by instead describing a utility function. This function assigns a real number to each bundle, and the agent selects the bundle with the highest utility.

In this paper, we focus on these "rationalizable" preferences, where agent choice can be implemented as utility maximization constrained by prices and income. The solution to these consumer choice problems provides us with, among other things, individual demand functions, which describe the choice of each good or service as a function of prices and income. The individual demand functions for each good are essential when aggregating to the market demand in Consumer Goods Market Aggregation, which in turn is used to find the price in a non-strategic equilibrium. In addition, we test variations on the framework such as the agents ability to make tradeoffs between the quantity of goods they would need to be able to purchase for an increase in the amount of work they provide for a given wage (i.e., the elasticity of labor supply), as well as cases of choice under uncertainty where the agent is choosing between possible lotteries under rationalizability assumptions required for von Neuman expected utility.

#### A.1.1 PROPERTIES OF UTILITY FUNCTIONS

In this section, we test the ability of the agent to use utility functions as a means to compare preferences over different "bundles" of goods or services. A key feature of economic reasoning in this context is for agents to consider how substitution between different goods in a bundle might achieve the same utility (i.e., map out the "indifference curves"). Key tests include correctly distinguishing between substitutes and complements in consumption, and calculating the marginal rate of substitution at a point on an indifference curve. This logic is essential for both agents acting as a planner as we will see in Appendix A.4 and when fulfilling the role of choice under budget and income constraints, in Deriving Demand.

**Element A.1 (Marginal Utility).** *The ability to calculate marginal utility for different types of demand curves such as quasilinear, Cobb-Douglas, and Leontief.*

**Element A.2 (Diminishing Marginal Utility).** *The ability to recognize the role of diminishing marginal utility in consumption decisions and the role of achieving interior solutions.*

**Element A.3 (Marginal Rate of Substitution).** *The ability to calculate the marginal rate of substitution between two goods in a consumption decision.*

**Element A.4 (Tangency and the Marginal Rate of Substitution).** *The ability to calculate the marginal rate of substitution between two goods in a consumption decision at a given point in the budget constraint as tangent to the indifference curve.*

**Element A.5 (Substitutes and Complements).** *The ability to distinguish between substitutes and complements in consumption decisions.*

#### A.1.2 DERIVING DEMAND

The module in this section tests an agent's ability to solve a constrained utility maximization problem to derive a demand function—relying on the results of Properties of Utility Functions. We test the canonical classes of demand functions, check the duality of Marshallian demand and Hicksian demand, and ask the agent to derive these demand functions from first principles.

**Element A.6 (Derivation of Marshallian Demand).** *The ability to calculate the demand curve for a good given a utility function and a budget constraint.*

**Element A.7 (Derivation of Hicksian Demand from Expenditure Minimization).** *The ability to calculate the demand curve for a good given a utility function and a budget constraint.*

**Element A.8 (Duality of Hicksian Demand).** *The ability to recognize that Hicksian demand (expenditure minimization) is dual to maximization in Marshallian Demand.*

### A.1.3  COMPARATIVE STATICS OF DEMAND

This module considers how agents reason about changes in prices or income, and their effects on the quantity of each good they would purchase. We test the classic law of demand, different types of goods (e.g., normal, inferior, and Giffen), and derive Engel curves from first principles. The key tests are to ensure the agent rationally responds to changes in relative prices, and investigate their substitution between goods in a bundle. In practice, these tests involve comparative statics of the $\arg\max$ from the utility maximization of the previous section on Deriving Demand—i.e., using an Envelope theorem and perturbing prices or income.

**Element A.9 (Law of Demand).** *The ability to calculate the change in demand with the change in price for normal goods.*

**Element A.10 (Price Elasticity of Demand).** *The ability to calculate the price elasticity of demand for a good given a utility function and a budget constraint.*

**Element A.11 (Consumption Changes).** *The ability to change the relative expenditures on goods given changes in relative prices with ordinary or Giffen goods.*

**Element A.12 (Engel Curves).** *The ability to calculate the Engel curve for a good given a utility function and a budget constraint.*

**Element A.13 (Income Elasticity of Demand).** *The ability to calculate the income elasticity of demand for a good given a utility function and a budget constraint.*

### A.1.4  LABOR SUPPLY

While the proceeding elements tested tradeoffs in choices of bundles with different goods, services (in Deriving Demand and over lotteries in Dynamic Consumption Decisions), often agents need to make a choice trading off between leisure and consumption. The elements in this module test an agent's ability to optimally make that tradeoff by balancing the consumption goods required to compensate for decreased leisure—which leads to the labor supply elasticity central to many branches of economics. Since goods must be purchased, agents will consider the relative wage from additional work compared to the price of goods. This leads us to be able to test an agent's ability to distinguish real from nominal prices.

**Element A.14 (Deriving Labor Supply).** *The ability to calculate the labor supply curve given specific preference parameterizations such as separable preferences or homothetic preferences.*

**Element A.15 (Labor Supply Elasticity).** *The ability to calculate the elasticity of labor supply.*

**Element A.16 (Marginal Rate of Substitution in Labor Supply).** *The ability to calculate the marginal rate of substitution between consumption and leisure in a labor supply decision.*

### A.1.5  DYNAMIC CONSUMPTION DECISIONS

Individuals often face decisions about how to trade off more consumption today at the cost of additional debt and less consumption in the future, and how best to plan for consumption with various contingencies with the future is uncertain. Among other applications, this provides a formal model of how to best choose a mixture of financial assets—i.e., portfolios. Consequently, this subsection tests intertemporal consumption choices, optimal portfolio choice—which involves selecting a mix of assets that maximizes expected utility given the risks and returns associated with each asset. Understanding portfolio choice helps explain how consumers manage risk and make investment decisions, which is vital for financial planning and economic stability.

**Element A.17 (Price of Risk with Mean-Variance Utility).** *The ability to calculate the price of risk for a mean-variance utility function.*

**Element A.18 (State-Contingent Consumption).** *The ability to calculate the optimal consumption given a utility function and a set of state-contingent consumption bundles.*

**Element A.19 (Arbitrage).** *The ability to recognize and execute arbitrage opportunities given two goods and prices you can resell.*

**Element A.20 (Optimal Portfolio Choice with Bid-Ask Spreads).** *The ability to calculate the optimal portfolio given bid-ask spreads.*

**Element A.21 (Exponential Discounting).** *The ability to exponentially discount future rewards or costs.*

**Element A.22 (Intertemporal Consumption Smoothing).** *The ability to calculate a smoothed consumption path and determine whether it is preferred to a non-smoothed path.*

## A.2 DECISIONS ON PRODUCTION IN NON-STRATEGIC ENVIRONMENTS

In the previous section, we derived how an agent facing a set of prices would choose the quantity demanded of each good or service to maximize their utility function. We also tested the amount of time that an agent might choose to wok (i.e., the quantity of labor supplied) given market wages—where the agent trades off the additional goods they might purchase against the lost leisure time they must forgo. Here, we look at the other side of the market and test an agent's ability to operate a production technology to maximize profits. Facing market prices for all production factors (e.g., wages and the capital) and the market price of the good or service they produce, the agent chooses the quantity of each factor of production and the total output. Parallel to DECISIONS ON CONSUMPTION IN NON-STRATEGIC ENVIRONMENTS, in Properties of Production Functions we first test general properties of production functions to ensure the agent can reason about substitution between factors, economies of scale in production, etc. Then in Deriving Factor Demand we solve the firms optimal profit maximization problem to determine the optimal choice of factors of production and output given a set of market prices. Finally, in Comparative Statics with Production we test the agents ability to reason about comparative statics on prices and their impact on factor demand and firm output.

### A.2.1 PROPERTIES OF PRODUCTION FUNCTIONS

Production functions in these environments take continuous inputs of each factor, which lets us test an agent's ability to conduct marginal thinking when choosing the composition of inputs. For example, by knowing the hourly wage of hiring an additional worker, the additional output the worker might produce using the particular production process, and the price they can sell the firm's output, they can decide whether hiring the additional worker is profitable. In the absence of prices, this section tests basic decision making of the agent for understanding substitution between factor of production, marginal products for each input, and the understanding of the returns to scale of a production process.

**Element A.23 (Marginal Products).** *The ability to calculate separate marginal products for a production function with multiple inputs (e.g., labor and capital).*

**Element A.24 (Input Price Elasticity).** *The ability to calculate the responsiveness of output to a proportional change in a specific input's cost, holding all other inputs constant.*

**Element A.25 (Output Elasticity).** *The ability to calculate the output elasticity of an input in a production function.*

**Element A.26 (Elasticity of Substitution).** *The ability to calculate the marginal elasticity of substitution between inputs in a production function.*

**Element A.27 (Diminishing Marginal Products).** *The ability to calculate the diminishing marginal products for a production function with multiple inputs.*

**Element A.28 (Average and Marginal Costs).** *The ability to calculate average and marginal costs given a production function and input prices, and use them to determine scale.*

**Element A.29 (Returns to Scale).** *The ability to determine the proportional change in output resulting from a proportional change in all inputs in a production function.*

### A.2.2 DERIVING FACTOR DEMAND

This module tests the agent's ability to act in the role of a profit maximizer in non-strategic situations where they take as given the price which they could sell goods they produce, and must pay for inputs to their production process at market rates (e.g., a competitive wage). Whereas in Deriving Demand, the agent was solving a utility maximization problem subject to a budget constraint, here they solve a profit maximization problem constrained by a production function. We test decisions on the quantity and composition of inputs, and the quantity of output for canonical production functions such as Cobb-Douglas and Leontief production functions given the agent's understanding of production

functions from Properties of Production Functions. The agent is asked to derive the factor demand functions from first principles from profit maximization and test their ability to reason with the dual cost-minimization formulation—analogous to the Hicksian vs. Marshallian demand of Deriving Demand.

**Element A.30 (Profit Maximization).** *The ability to calculate the optimal input bundle for a firm given a production function and input prices. Examples of given production functions: Cobb-Douglas, Leontief, Perfect Substitutes, CES production, CRS production, fixed costs.*

**Element A.31 (Expenditure Minimization).** *The ability to calculate the optimal input bundle for a firm given a production function and input prices.*

**Element A.32 (Duality of Profit Maximization and Expenditure Minimization).** *The ability to recognize that profit maximization is dual to expenditure minimization in production decisions and achieve consistent solutions.*

### A.2.3 COMPARATIVE STATICS WITH PRODUCTION

This module considers how agents reason about changes in the prices at which they can sell their goods, as well as changes in the costs of producing those goods. In particular, we can test how this affects their optimal choice of inputs to their production process (e.g., how many people to hire or robots to lease). We test comparative statics on the prices of inputs to the production function, changes to the underlying production technology, and substitution between goods for classic production functions such as Cobb-Douglas and Leontief. Analogous to the relationship between Deriving Demand and Comparative Statics of Demand, these tests involve comparative statics of the $\mathrm{argmax}$ from the profit maximization of Deriving Factor Demand—i.e., using an Envelope theorem and perturbing factor prices.

**Element A.33 (Price Elasticity of Supply).** *The ability to calculate the price elasticity of supply for a good given a production function and input prices.*

**Element A.34 (Shephard's Lemma).** *The ability to calculate factor demands given a cost function using the derivatives with respect to prices.*

**Element A.35 (Input Price Elasticity).** *The ability to calculate how the optimal input bundle changes with changes in input prices for a given production function.*

**Element A.36 (Total Factor Productivity).** *The ability to calculate total factor productivity given a production function and input prices*

### A.2.4 DYNAMIC PRODUCTION DECISIONS

While Deriving Factor Demand tested the ability of agents to make static (i.e., within-period) decisions on the mix of input factors to maximize profits, many producer problems are inherently dynamic. For example, we can test if an agent can optimally choose the amount of capital to purchase given forecasts of future consumer demand and prices or choose how much to adjust the labor force in cases when labor is difficult to relocate due to frictions such as hiring and firing costs. Finally, agents are tested on their ability to make optimal entry and exit decisions based on their forecasted profits in an evolving market.

**Element A.37 (Dynamic Profit Maximization).** *The ability to calculate the optimal investment decision given a production function and input prices.*

**Element A.38 (Entry and Exit Decisions).** *The ability to calculate the optimal entry and exit decisions given a production function and fixed costs.*

### A.3 DECISIONS IN MULTI-AGENT NON-STRATEGIC ENVIRONMENTS

This setting tests the core logic of the relationship between supply-and-demand and prices, building on the tests of optimal behavior in appendix A.2 and appendix A.1. Economists refer to "general equilibrium" as the process where equilibrium prices and quantities emerge with a large number of non-strategic, price-taking market participants interact. Unlike the strategic models found in STEER, the assumption is that the market interactions that lead to this equilibrium occur through an unspecified process that clears markets (i.e., a "Walrasian auctioneer" or "invisible hand").

In particular, for non-strategic settings, all market participants take prices as given and choose the quantity demanded or supplied in each market. For example, consumers jointly decide on the quantity demanded of goods and services given relative prices, and the quantity of labor supplied given a wage. Simultaneously, producers choose the quantity supplied of the good and the demand of each factor of production. With a large number of non-strategic market participants we can test the agents ability aggregate all of their supply and demand functions to calculate a market-level supply and demand. Finally, given the aggregated supply and demand functions for each market, we can test whether an agent can find the market clearing price where supply is equal to demand in equilibrium—given their internal model of all the market participants.

In this section, we organize by markets rather than by the role of a decision maker, as in the previous sections. For example, in the goods market we first ensure agents understand how individual demand functions from Deriving Demand aggregate to a market demand function for the good given a price, then that the agent understands how to aggregate the output from each producer at a given price from Comparative Statics with Production, and finally that the agent is able to calculate the price which would equate demand and supply and clear the market in a non-strategic setting. Factor markets are treated similarly.

Finally, given a system of equations that defines an equilibrium price we can perturb primitives (e.g., technological factors, distortions on decisions such as tax rates, or exogenous prices not determined in equilibrium) to see how the market clearing price would respond. That is an essential tool for agents to be able to reason about the impact of interventions and distortions in Appendix A.4.

### A.3.1 Consumer Goods Market Aggregation

The market clearing prices in general equilibrium arise from the separate market-level demand and supply curves, which sums the demand or supply across all market participants at a given price. Here we test the aggregation of demand functions derived from individual preferences, as in Deriving Demand and Comparative Statics of Demand, to a market demand function that summarizes the total quantity demanded across all agents at a given price. Central to the tests is to verify that the agent can aggregate the demands of market participants with heterogeneous preferences. On the other side of the market, we test if the agent can aggregate the "supply functions" resulting from the optimal choice of factors in Deriving Factor Demand and Comparative Statics with Production.

**Element A.39 (Aggregation of Consumer Demand).** *The ability to calculate the aggregate demand for a good given primitives of demand into expenditure shares.*

**Element A.40 (Aggregation of Offer Curve for the Good).** *The ability to calculate the aggregate supply of a good given primitives of supply into production functions.*

### A.3.2 Factor Market Aggregation

As with the case of the goods market in Consumer Goods Market Aggregation the market demand and supply for factors of production are essential to find the market clearing price. For example, we test whether the agent can aggregate the individual labor supply curve decisions from market participants who work at a particular wage, following Labor Supply, into a market labor supply curve. On the other side of the market, we test whether the agent can aggregate the labor demand in Deriving Factor Demand from producers into a market labor demand curve. The same tests are essential for all factors of production, including capital.

**Element A.41 (Aggregation of Labor Demand).** *The ability to calculate the aggregate demand for labor given primitives of demand into expenditure shares.*

**Element A.42 (Aggregation of Capital Demand).** *The ability to calculate the aggregate demand for capital given primitives of demand into expenditure shares.*

**Element A.43 (Aggregation of Labor Supply).** *The ability to calculate the aggregate supply of labor given primitives of supply into production functions.*

**Element A.44 (Aggregation of Fixed Factor Supply).** *The ability to calculate the aggregate supply of capital given primitives of supply into production functions.*

### A.3.3 PRICES IN STATIC MARKET EQUILIBRIUM

In this setting we test the agent's ability to reason about how prices emerge in non-strategic setting as a process of equating supply and demand, which in turn relies on their ability to aggregate those market demand functions from consumer and producer behavior.

More specifically, the core logic of general equilibrium is to find the equilibrium price by taking the aggregated demand and supply functions for each market and find the prices which would equate demand and supply. For example, the supply and demand functions for the good, as a function of the price, in Consumer Goods Market Aggregation, or the supply and demand functions for factors of production, as a function of factor prices in Factor Market Aggregation. This is done market by market, taking all other prices as given—which requires the agent reason through comparative statics of the solution to a system of equations while keeping everything else fixed.

**Element A.45 (Find Equilibrium Price).** *The ability to calculate the equilibrium prices given a production function and a demand function.*

**Element A.46 (Factor Shares in Equilibrium).** *The ability to calculate the factor shares in a competitive equilibrium given a production function and input prices.*

### A.3.4 COMPARATIVE STATICS OF EQUILIBRIUM PRICES

Here, we test whether agents can reason about how prices and allocations (e.g., labor, capital, and goods) would respond to changes in the environment. The canonical tests are to see how changes in model primitives (e.g., productivity of the production process) or exogenous forces from outside the model (e.g., impact of weather), change the equilibrium price and allocations of labor, capital, etc. that would clear the market and equate demand and supply.

**Element A.47 (Comparative Statics with Total Factor Production Shocks).** *The ability to calculate how equilibrium prices change with changes in input prices for a Cobb-Douglas production function.*

**Element A.48 (Comparative Statics with Inelastic or Perfectly Elastic Supply).** *The ability to calculate how equilibrium prices change with changes in input prices for a production function with inelastic or perfectly elastic supply.*

### A.4 EVALUATING EQUILIBRIA AND EXTERNALITIES

In DECISIONS IN MULTI-AGENT NON-STRATEGIC ENVIRONMENTS, we tested an agents ability to reason about equilibrium prices and quantities arising from supply and demand decisions in a non-strategic setting. Although preferences were reflected in the underlying supply and demand functions themselves (i.e., utility maximization in the consumption decisions of DECISIONS ON CONSUMPTION IN NON-STRATEGIC ENVIRONMENTS and profit maximization in the production decisions of DECISIONS ON PRODUCTION IN NON-STRATEGIC ENVIRONMENTS), the equilibria in DECISIONS IN MULTI-AGENT NON-STRATEGIC ENVIRONMENTS do not necessarily reflect broader social preferences.

However, we can still ask whether the resulting "allocations" (i.e., the physical goods produced and how they are distributed to individuals, the amount of hours worked, and the physical capital installed) from the "invisible hand" in DECISIONS IN MULTI-AGENT NON-STRATEGIC ENVIRONMENTS compare to a alternative ways of allocating resources which may directly take social preferences into account. A central result of economics in non-strategic settings is that absent market imperfections and market power (i.e., when self-interested agents cannot directly manipulate prices because they are too small) the competitive equilibria of DECISIONS IN MULTI-AGENT NON-STRATEGIC ENVIRONMENTS typically yields the same allocations a benevolent planner might choose.

In this section, we consider how a social planner would evaluate the underlying welfare, efficiency, and inequality that comes about in non-strategic equilbria with prices derived from equating supply and demand. This leads to testing the ability of the agent to evaluate Pareto efficiency, consider the welfare theorems, evaluate Pigouvian externalities, and weigh the welfare impact of various market interventions which change the equilibria derived in DECISIONS IN MULTI-AGENT NON-STRATEGIC ENVIRONMENTS.

### A.4.1 WELFARE AND DECENTRALIZATION

In this section, we test whether the agent can determine cases where the the competitive equilibrium they calculate would yield the same distribution of resources and consumer welfare as that of a benevolent social planner directly making the consumption and production decisions of all agents directly (also known as the "Welfare Theorems"). In cases where the supply-and-demand relationships lead to the same results as those of a planner, the competitive equilibrium and its prices are said to "decentralize" the problem of a social planner. We then test that the agent recognizes cases where the welfare theorems fail, and can calculate the degree of welfare loss due to the distortions.

**Element A.49 (First Welfare Theorem).** *The ability to recognize that a competitive equilibrium is Pareto efficient.*

**Element A.50 (Second Welfare Theorem).** *The ability to recognize that any Pareto efficient allocation can be achieved as a competitive equilibrium with prices.*

**Element A.51 (Consumer Surplus).** *The ability to calculate consumer surplus given a demand curve and a price.*

**Element A.52 (Producer Surplus).** *The ability to calculate producer surplus given a supply curve and a price.*

**Element A.53 (EFficient Surplus).** *The ability to calculate the total surplus in a competitive equilibrium and recognize that it is maximized in the competitive equilibrium.*

**Element A.54 (Deadweight Loss of a Monopoly).** *The ability to calculate the deadweight loss of a monopoly given a demand curve and a supply curve.*

### A.4.2 WELFARE ANALYSIS OF MARKET EQUILIBRIUM

In this section, we focus on the agent's ability to evaluate welfare implications of various forms of market equilibrium, particularly how different policies and distortions impact overall efficiency and resource allocation. The agent is tested on their understanding of how different interventions—such as taxes, subsidies, and price controls—affect welfare outcomes, and their ability to distinguish between distortionary and non-distortionary policies.

**Element A.55 (Identify Non-Distortionary Taxes).** *The ability to identify taxes which do not distort the allocation of resources.*

**Element A.56 (Irrelevance of Tax Incidence).** *The ability to recognize that the incidence of a tax does not depend on who is legally responsible for paying the tax.*

**Element A.57 (Labor Supply Distortions).** *The ability to determine the extent that labor taxes will distort labor supply and change aggregates and prices.*

**Element A.58 (Capital Market Distortions).** *The ability to identify that taxing a fixed factor is non-distortionary, but distorts with dynamic accumulation.*

## B MITIGATING DATA CONTAMINATION WITH AUTO−STEER

Data contamination, where training data inadvertently includes information from test sets, poses significant challenges in machine learning, leading to overestimated model performance and compromised generalization capabilities. To address this, we implemented a structured dataset generation methodology incorporating human oversight, controlled data generation, and style transfer techniques. This appendix details our approach and its alignment with best practices in the literature.

The `auto-STEER` methodology provides a systematic approach to generating datasets that mitigates the risk of data contamination, ensuring the integrity of benchmarks and the validity of results. Below, we outline the key aspects of `auto-STEER` that address this issue:

**Challenging Models with Rephrasings:**

Rephrasings are known to cause significant variance in model performance, as demonstrated in the GSM-Symbolic dataset Mirzadeh et al. (2024) and other studies (e.g., Zhu et al., 2024; Wang et al., 2023) highlighting how syntactic or stylistic changes can challenge generalization. In Appendix G, we also show that much of the observed variance in LLM performance arises from these rephrasings,

underscoring their role in robust evaluations. `auto-STEER` leverages this phenomenon to craft diverse rephrased questions that test beyond rote learning.

**Systematic Question Generation:**

`auto-STEER` generates new questions through a structured process that balances diversity and consistency. Questions are systematically rephrased or style-transferred to ensure they are different enough from the original templates to prevent memorization while retaining the same core meaning. This approach reduces the risk of overlap with pre-trained data while preserving the focus of the assessment.

The rapid advancement of large language models necessitates benchmarks that can evolve just as quickly. To address this, `auto-STEER` incorporates a user interface (UI) that allows users to regenerate entire datasets with minimal effort. By modifying domains, seeds, or even resampling numerical values, users can quickly produce a new dataset tailored to the latest needs. This adaptability ensures that benchmarks remain fresh and resistant to contamination as models advance.

Through these features, `auto-STEER` provides a robust mechanism for creating datasets that challenge models in meaningful ways while maintaining a high degree of control over data integrity. The systematic generation of diverse, rephrased questions and the ability to regenerate datasets on demand make it a powerful tool for addressing data contamination in an era of rapidly evolving AI capabilities.

## C   TECHNICAL DESCRIPTIONS OF ADAPTATIONS

### C.1   RPM (CONDITIONING):

Given the LLM's output distribution over all possible tokens, filter to include only those that correspond to valid options. For example, if a question has four options then get the probabilities corresponding to 'A', 'B', 'C', and 'D'. Then, compute softmax over the valid options to normalize the filtered probabilities into a distribution.

### C.2   RPM (MIXING):

Alternatively, we restrict the output distribution to only valid option tokens $O$ as follows: $\alpha \cdot p(o) + (1 - \alpha)1/|O|$, where $o \in O$, $p(o)$ is the probability the LLM assigns to each token it outputs, and $\alpha = \sum_{o \in O} p(o)$. We then compute the softmax to normalize the resulting probabilities into a distribution.

In the mixing approach, if an LLM is confident in a valid option token the resulting distribution will place high probability on that token, but if an LLM places negligible probability on the valid option tokens then the resulting distribution will more closely resemble a uniform distribution.

### C.3   PROMPT FOR GETTING ANSWERS FROM FREE-TEXT QA

**User Message:**

Report the answer that the following reasoning ended up with. Do not solve the question just look at the explanation text and report the answer if it exists or None if not. Just report the number or None.
Q: {question_text}
Explanation: {model_response}

## D   TECHNICAL DESCRIPTIONS OF METRICS

### D.1   ACCURACY.

Accuracy is the most broadly used metric for evaluating LLMs. We define accuracy metrics as metrics that only look at the top token that the LLM outputs.

### D.1.1 Exact-match accuracy

This is the fraction of questions answered correctly.

### D.1.2 Normalized accuracy

Elements can differ in their number of multiple choice options, leading to differences in the exact-match accuracy of random guessing. We can compensate for this by reporting the gap between the LLM's exact-match accuracy and random guessing (Budescu & Bar-Hillel, 1993). We compute normalized accuracy for an element as follows: $\sum_{i=1}^{N} a_i(t_i) - \frac{1-a_i(t_i)}{|O_i|-1}$, where $t_i$ is the top token the LLM outputs for question $i$, $a_i$ is the indicator describing whether the top token is correct or not, $N$ the number of questions in the element, and $|O_i|$ the number of options in the question. In other words, normalized accuracy rewards an LLM with 1 point for every correct answer and penalizes an LLM by 1 over the number of options minus 1 for each incorrect answer.

## D.2 Calibration

It can also be useful to understand how confident an LLM is in its responses and the extent to which these confidence levels align with accuracy.

### D.2.1 Expected calibration error

We follow Liang et al. (2022) and Raman et al. (2024) in measuring the confidence of an LLM's response and computing the expected calibration error (ECE; Naeini et al., 2015; Guo et al., 2017). ECE measures how closely the probability an LLM assigns to its top answer matches the actual probability of the correct answer, which in our case is 1. ECE first splits the data into $M$ equally spaced bins, where each bin contains the probabilities the model assigned to their top token in that range: e.g., let $p^{\max}$ be the set of most probable tokens for each question then if $M = 2$, then the first bin $B_1 = \{p \mid p \in p^{\max} \text{ and } p \in [0, 0.5]\}$. It is then defined as $\sum_{i \in [M]} |B_i|/N \cdot |\mathtt{acc}(B_i) - \mathtt{conf}(B_i)|$, where $\mathtt{conf}(B_i)$ is the average probability the LLM assigned to its top token in bin $B_i$, and $\mathtt{acc}(B_i)$ denotes the exact-match accuracy in bin $B_i$. We allow users to choose the number of bins, however, we set $M = 10$ uniformly spaced over the interval $[0, 1]$ as is standard.

### D.2.2 Brier Score

The Brier Score of an element is defined as

$$\sum_{i=1}^{N} \frac{1}{|O_i|} \sum_{o \in O_i} (p_i(o) - a_i(o))^2,$$

where $p_i(o)$ is the probability the LLM assigns to option $o$ in question $i$. Thus, if an LLM is overly confident in an incorrect answer (e.g., assigns a probability of 0.9 to a wrong option), the Brier Score will penalize it more heavily.

### D.2.3 Expected Probability Assignment

EPA measures how much probability mass an LLM assigns to the correct answer option out of all possible options. It is defined as: $1/N \sum_{i \in [N]} p_i^*$, where $p_i^*$ is the predicted probability that the LLM assigns to the correct option for question $i$.

## D.3 Robustness

Elements are not the lowest level of granularity in our benchmark and aggregating on an element-by-element basis may hide where LLM performance varies. We introduce three robustness metrics that aggregate on the component fields of an element: domains, types, and perspectives. We compute the domain (type, perspective) robustness on each element by taking the minimum normalized accuracy over all domains (types, perspectives).

# E   EXAMPLES

## E.1   GPT-4O AGGREGATION OF CONSUMER DEMAND (SHOWN)

**User Message:**

Q: As the owner of a premium bedding brand, I cater to a diverse group of customers. I've discovered that there are 1204 potential buyers for my luxury bedding products. Out of these, 820 shoppers in urban markets each follow the demand function $18.1 - 1.51P$, while 384 shoppers in suburban markets each adhere to the demand function $75.44 - 8.68P$. How can I calculate the total demand for my bedding at a price point of 8.4?
A.  2779.36
B.  1832.01
C.  5411.87 *(Correct Answer)*
D.  5050.03
Let's think step by step. Explain your reasoning.

---

**GPT-4o Message:**

To calculate the total demand for your luxury bedding products at a price point of 8.4, we need to evaluate the demand functions for both urban and suburban markets separately and then sum the results.
**Step 1: Calculate the demand in the urban market**
The demand function for the urban market is given by:

$$Q_u = 18.1 - 1.51P$$

Substitute $P = 8.4$ into the urban demand function:

$$Q_u = 18.1 - 1.51 \times 8.4$$

$$Q_u = 18.1 - 12.684$$

$$Q_u = 5.416$$

Since there are 820 shoppers in the urban market, the total demand in the urban market is:

$$\text{Total Urban Demand} = 820 \times 5.416$$

$$\text{Total Urban Demand} = 4441.12$$

**Step 2: Calculate the demand in the suburban market**
The demand function for the suburban market is given by:

$$Q_s = 75.44 - 8.68P$$

Substitute $P = 8.4$ into the suburban demand function:

$$Q_s = 75.44 - 8.68 \times 8.4$$

$$Q_s = 75.44 - 72.912$$

$$Q_s = 2.528$$

Since there are 384 shoppers in the suburban market, the total demand in the suburban market is:

$$\text{Total Suburban Demand} = 384 \times 2.528$$

$$\text{Total Suburban Demand} = 970.88$$

**Step 3: Calculate the total demand**
Now, sum the total demand from both markets:

$$\text{Total Demand} = \text{Total Urban Demand} + \text{Total Suburban Demand}$$

$$\text{Total Demand} = 4441.12 + 970.88$$

$$\text{Total Demand} = 5412$$

Therefore, the total demand for your bedding at a price point of 8.4 is approximately 5411.87.
The correct answer is:
C. 5411.87

## E.2   GPT-4o Aggregation of Consumer Demand (Hidden)

**User Message:**

Q: As the owner of a premium bedding brand, I cater to a diverse group of customers. I've discovered that there are 1204 potential buyers for my luxury bedding products. Out of these, 820 shoppers in urban markets each follow the demand function $18.1 - 1.51P$, while 384 shoppers in suburban markets each adhere to the demand function $75.44 - 8.68P$. How can I calculate the total demand for my bedding at a price point of 8.4?
Let's think step by step. Explain your reasoning.

**GPT-4o Message:**

To calculate the total demand for your luxury bedding at a price point of 8.4, we need to evaluate the demand functions for both urban and suburban markets separately and then sum the results. Here's a step-by-step breakdown:
1. **Identify the demand functions:**
- Urban market demand function: $Q_u = 18.1 - 1.51P$
- Suburban market demand function: $Q_s = 75.44 - 8.68P$
2. **Substitute the price point (P = 8.4) into each demand function:**
- For the urban market:
$$Q_u = 18.1 - 1.51 \times 8.4$$
Calculate the value:
$$Q_u = 18.1 - 1.51 \times 8.4 = 18.1 - 12.684 = 5.416$$
- For the suburban market:
$$Q_s = 75.44 - 8.68 \times 8.4$$
Calculate the value:
$$Q_s = 75.44 - 8.68 \times 8.4 = 75.44 - 72.912 = 2.528$$
3. **Sum the demands from both markets to get the total demand:**
$$Q_{total} = Q_u + Q_s = 5.416 + 2.528 = 7.944$$

Therefore, at a price point of 8.4, the total demand for your luxury bedding products from both urban and suburban markets is approximately 7.944 units.

**User Message:**

A.  2779.36
B.  1832.01
C.  5411.87 *(Correct Answer)*
D.  5050.03
Answer by writing the option letter corresponding to the correct option. WRITE ONLY A SINGLE LETTER.
A:

**GPT-4o Message:**

B

## E.3 GPT-4O PROFIT MAXIMIZATION

**User Message:**

Q: John manages a company that produces premium leather wallets. The production output depends on the capital (K) he invests, represented by the function $3.97K^{0.45}$. At the moment, the wallets are sold at a price of 3.64, and John has an initial capital of $K_1 = 3.3$. He needs to decide how much additional capital to invest for the next day's market. John is aware of the potential price distribution for his wallets for the next day, represented by price 2.7 with probability 0.24, price 2.14 with probability 0.42, price 3.52 with probability 0.34. There is also a cost associated with increasing his capital, given by $(K_1 - K_2)^2$, and he applies a discount factor of 0.37 to future revenues. To maximize his profits, how much additional capital should John invest?
A. 0.44
B. 0.14
C. 1.1
D. 0.09
Let's think step by step. Explain your reasoning.

**GPT-4o Message:**

To determine how much additional capital John should invest to maximize his profits, we need to consider the expected revenue from the different potential prices, the cost of increasing capital, and the discount factor applied to future revenues. Let's break this down step by step.
**Step 1: Calculate the Expected Revenue**
First, we need to calculate the expected price of the wallets for the next day using the given probabilities.

$$\text{Expected Price} = (2.7 \times 0.24) + (2.14 \times 0.42) + (3.52 \times 0.34)$$

$$\text{Expected Price} = 0.648 + 0.8988 + 1.1968 = 2.7436$$

**Step 2: Determine the Production Function**
The production function is given by $3.97K^{0.45}$. We need to calculate the production output for different levels of capital $K$.
**Step 3: Calculate the Revenue for Different Capital Levels**
Let's denote the initial capital as $K_1 = 3.3$. We need to consider the additional capital $\Delta K$ that John might invest. The new capital $K_2$ will be $K_1 + \Delta K$.
The revenue function for a given capital $K$ is:

$$\text{Revenue} = \text{Expected Price} \times \text{Production Output}$$

$$\text{Revenue} = 2.7436 \times 3.97K^{0.45}$$

**Step 4: Calculate the Cost of Increasing Capital**
The cost of increasing capital is given by $(K_1 - K_2)^2$. Since $K_2 = K_1 + \Delta K$, the cost becomes:

$$\text{Cost} = (\Delta K)^2$$

**Step 5: Calculate the Discounted Future Revenue**
John applies a discount factor of 0.37 to future revenues. Therefore, the discounted future revenue is:

$$\text{Discounted Revenue} = 0.37 \times \text{Revenue}$$

**Step 6: Calculate the Net Profit**
The net profit is the discounted revenue minus the cost of increasing capital:

$$\text{Net Profit} = 0.37 \times (2.7436 \times 3.97K_2^{0.45}) - (\Delta K)^2$$

**GPT-4o Message (Continued):**

**Step 7: Evaluate the Net Profit for Different Values of $\Delta K$**
We need to evaluate the net profit for the given options of additional capital investment:
Option A: $\Delta K = 0.44$

$$K_2 = 3.3 + 0.44 = 3.74$$
$$\text{Production Output} = 3.97 \times 3.74^{0.45}$$
$$\text{Revenue} = 2.7436 \times 3.97 \times 3.74^{0.45}$$
$$\text{Discounted Revenue} = 0.37 \times \text{Revenue}$$
$$\text{Cost} = (0.44)^2 = 0.1936$$
$$\text{Net Profit} = 0.37 \times \text{Revenue} - 0.1936$$

Option B: $\Delta K = 0.14$

$$K_2 = 3.3 + 0.14 = 3.44$$
$$\text{Production Output} = 3.97 \times 3.44^{0.45}$$
$$\text{Revenue} = 2.7436 \times 3.97 \times 3.44^{0.45}$$
$$\text{Discounted Revenue} = 0.37 \times \text{Revenue}$$
$$\text{Cost} = (0.14)^2 = 0.0196$$
$$\text{Net Profit} = 0.37 \times \text{Revenue} - 0.0196$$

Option C: $\Delta K = 1.1$

$$K_2 = 3.3 + 1.1 = 4.4$$
$$\text{Production Output} = 3.97 \times 4.4^{0.45}$$
$$\text{Revenue} = 2.7436 \times 3.97 \times 4.4^{0.45}$$
$$\text{Discounted Revenue} = 0.37 \times \text{Revenue}$$
$$\text{Cost} = (1.1)^2 = 1.21$$
$$\text{Net Profit} = 0.37 \times \text{Revenue} - 1.21$$

Option D: $\Delta K = 0.09$

$$K_2 = 3.3 + 0.09 = 3.39$$
$$\text{Production Output} = 3.97 \times 3.39^{0.45}$$
$$\text{Revenue} = 2.7436 \times 3.97 \times 3.39^{0.45}$$
$$\text{Discounted Revenue} = 0.37 \times \text{Revenue}$$
$$\text{Cost} = (0.09)^2 = 0.0081$$
$$\text{Net Profit} = 0.37 \times \text{Revenue} - 0.0081$$

**Step 8: Compare the Net Profits**
We need to compare the net profits for each option to determine which one maximizes John's profit. After calculating the net profits for each option, we find that:
- Option A: $\Delta K = 0.44$ yields the highest net profit.
Therefore, the best option for John is to invest an additional capital of 0.44.
**Answer: A. 0.44**

**User Message:**

Answer by writing the option letter corresponding to the correct option. WRITE ONLY A SINGLE LETTER.
A:

**GPT-4o Message:**

A

# F    FIGURES

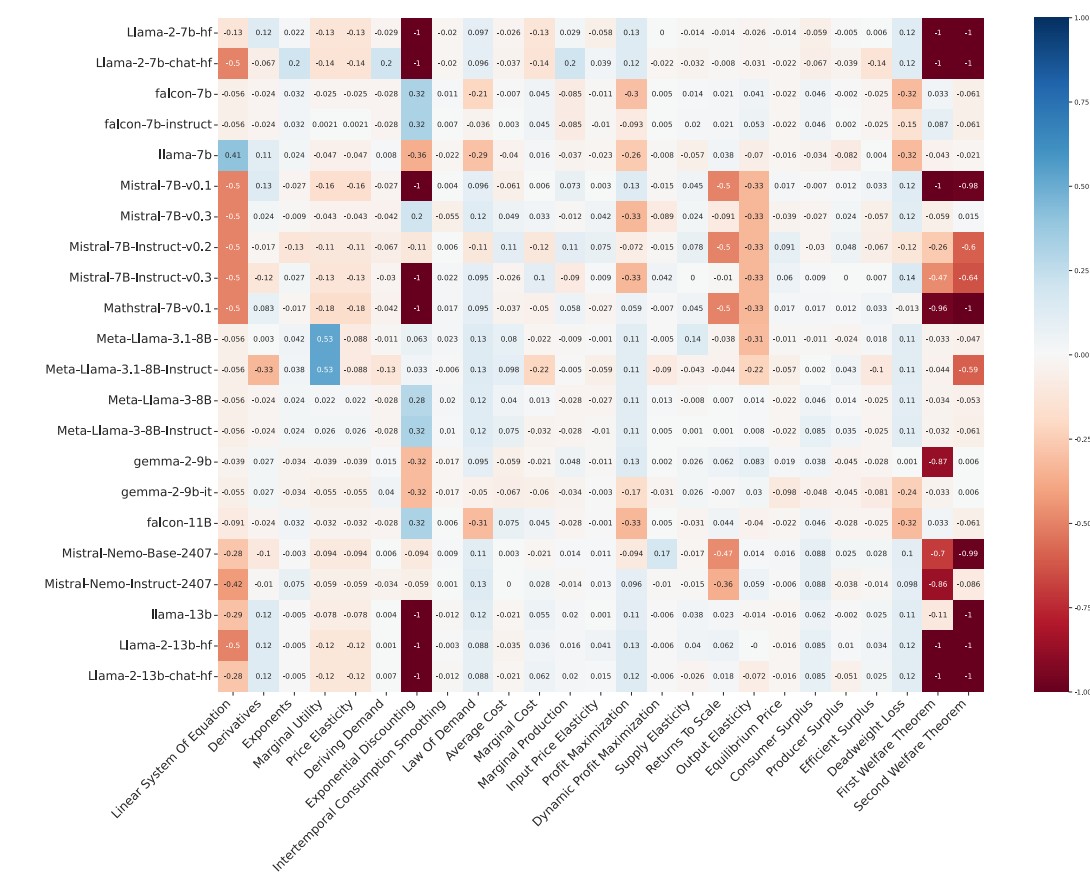

Figure 3: Heatmap of normalized accuracy of open-source models.

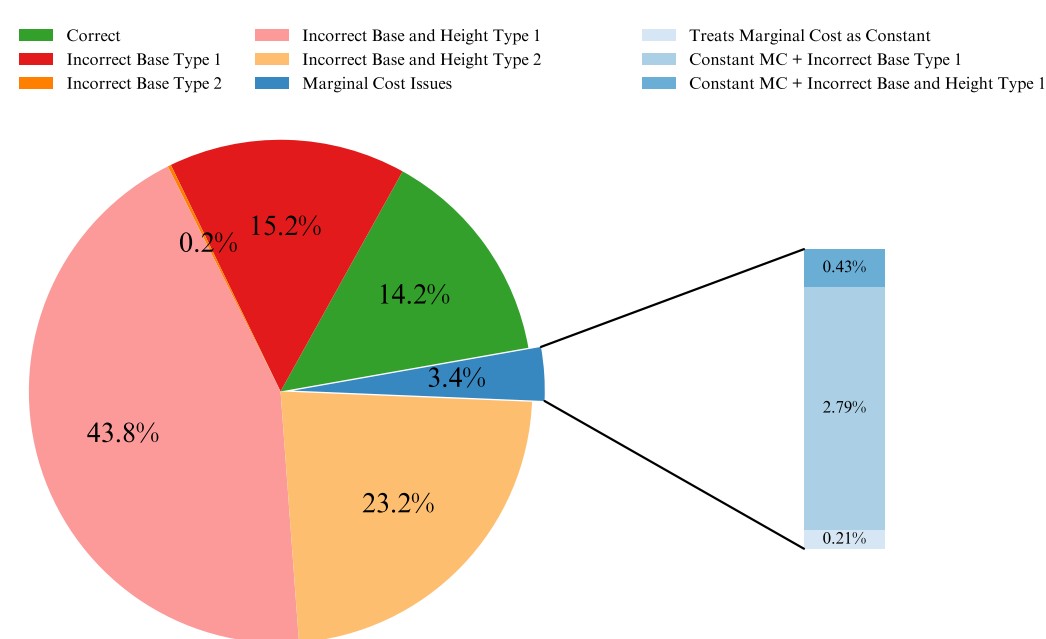

Figure 4: Error analysis of claude-3-5-sonnet on the Deadweight Loss of a Monopoly element. We further breakdown the errors by incorrectly interpreting the marginal cost.

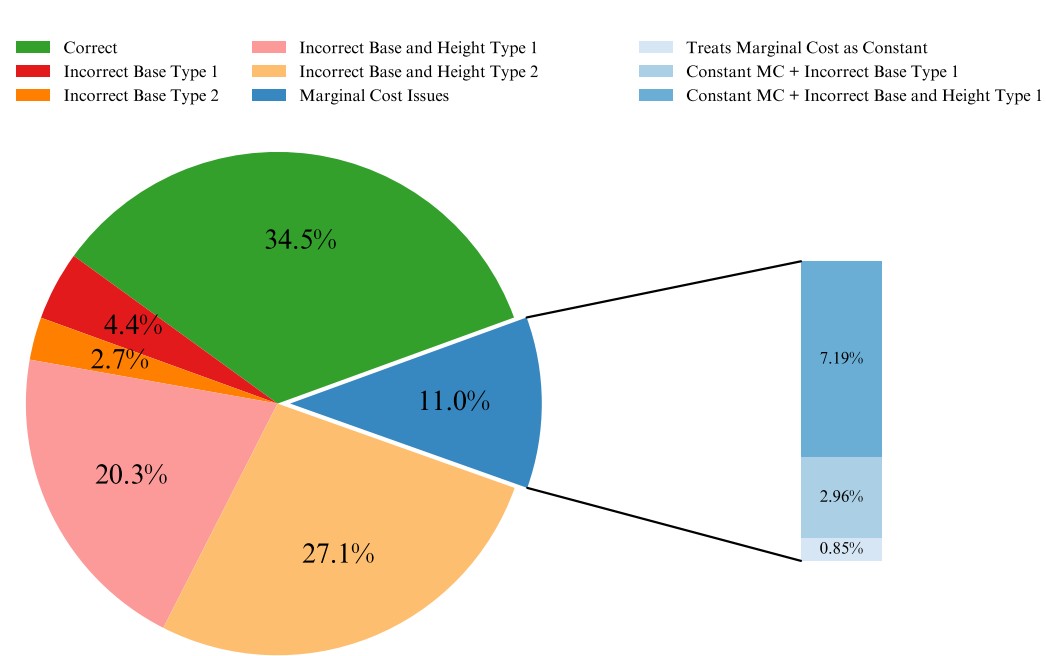

Figure 5: Error analysis of gpt-4o on the Deadweight Loss of a Monopoly element. We further breakdown the errors by incorrectly interpreting the marginal cost.

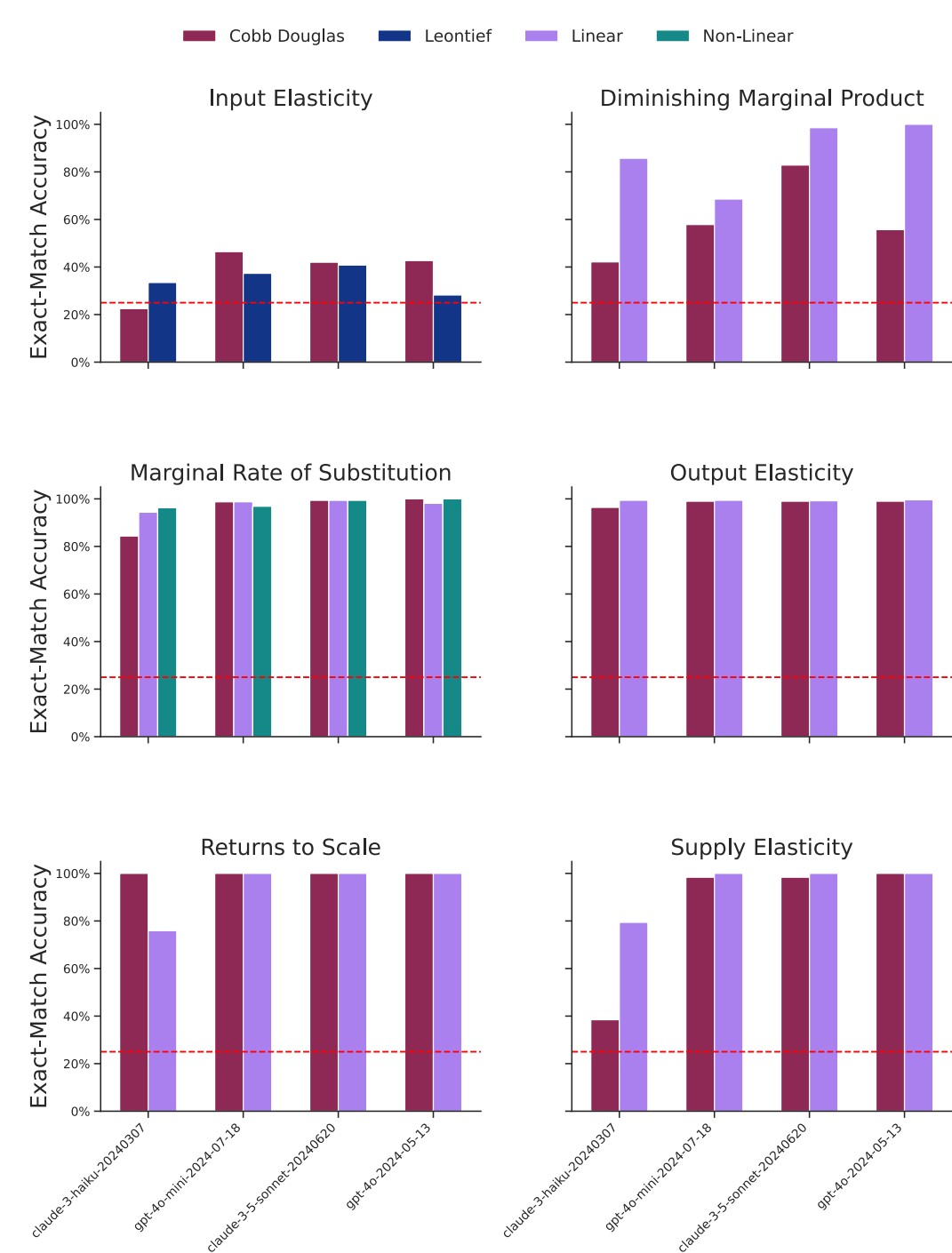

Figure 6: Exact-match accuracy of all closed-source models across six elements and four functional families (Cobb-Douglas, Leontief, Linear, and Non-Linear). The results demonstrate varying type robustness, with Cobb-Douglas being a generally more challenging functional family but not consistently harder for all elements. For instance, accuracy remains high for elements such as Output Elasticity and Marginal Rate of Substitution, even on the Cobb-Douglas functions, while elements like Input Price Elasticity and Returns to Scale show more variability across functional types. The red dashed line indicates the random guessing baseline for comparison.

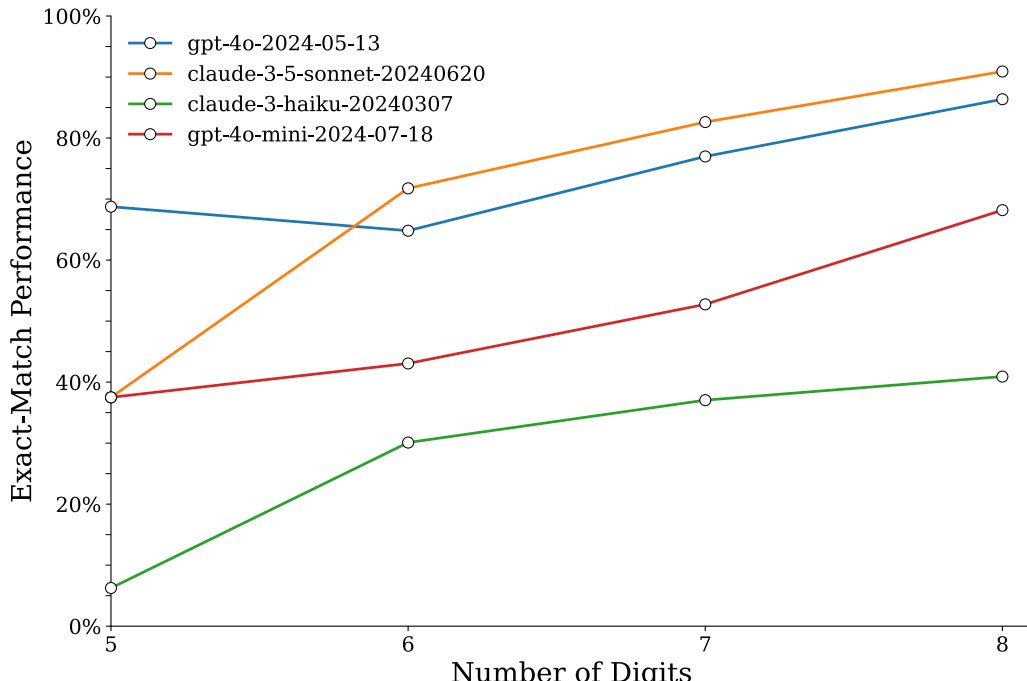

Figure 7: This figure depicts exact-match performance on the Aggregation of Consumer Demand element on the shown implementation of 0-CoT for the closed-source models against the number of digits of the correct answer.

# G  ANALYSIS OF REPHRASING VARIANCE

To understand the role of question rephrasings in our dataset, we conducted an analysis of variance (ANOVA) on all other controllable features. These features include type, domain, and perspective. The goal of this analysis was to quantify the variance in LLM performance attributable to these features and, by exclusion, infer the contribution of rephrasings to the remaining unexplained variance.

The results for the top-performing models, summarized in Table 2 through Table 6, indicate that the explained variance attributable to the controlled features is consistently low across all evaluated models. This leaves approximately 56% (for claude-3-5-sonnet) and up to 91% (for o1-preview) of the variance unexplained by the features included in the analysis. Given that question rephrasings are a systematic element of our dataset design and were not included as a feature in this analysis, we infer that the majority of this residual variance is due to differences in how models respond to semantically equivalent but syntactically varied prompts.

| Factor | Sum of Squares | Degrees of Freedom | F-Statistic | p-value |
|---|---|---|---|---|
| domain | 7.5572 | 11.0000 | 3.0199 | 0.0823 |
| perspective | 3.4351 | 5.0000 | 3.0199 | 0.0823 |
| CAR | 21.1235 | 1.0000 | 92.8524 | 0.0000 |
| element:type | 178.6251 | 260.0000 | 3.0199 | 0.0823 |
| Residual | 2218.5356 | 9752.0000 | | |
| | | *R-squared* | | 0.0941 |
| | | *Adjusted R-squared* | | 0.0907 |

Table 2: ANOVA Results for o1-preview-2024-09-12

| Factor | Sum of Squares | Degrees of Freedom | F-Statistic | p-value |
|---|---|---|---|---|
| domain | 0.2619 | 11.0000 | 0.0928 | 0.7607 |
| perspective | 0.1190 | 5.0000 | 0.0928 | 0.7607 |
| 0-CoT | 229.7295 | 1.0000 | 895.0635 | 0.0000 |
| CAR | 141.3015 | 1.0000 | 550.5338 | 0.0000 |
| element:type | 47.4945 | 1995.0000 | 0.0928 | 0.7607 |
| Residual | 16015.5015 | 62399.0000 | | |
| | | *R-squared* | | 0.3368 |
| | | *Adjusted R-squared* | | 0.3358 |

Table 3: ANOVA Results for gpt-4o-2024-05-13

| Factor | Sum of Squares | Degrees of Freedom | F-Statistic | p-value |
|---|---|---|---|---|
| domain | 0.4573 | 11.0000 | 0.1293 | 0.7191 |
| perspective | 0.2079 | 5.0000 | 0.1293 | 0.7191 |
| 0-Cot | 34.0324 | 1.0000 | 105.8911 | 0.0000 |
| CAR | 159.1899 | 1.0000 | 495.3161 | 0.0000 |
| element:type | 82.9338 | 1995.0000 | 0.1293 | 0.7191 |
| Residual | 18642.5829 | 58006.0000 | | |
| | | *R-squared* | | 0.2964 |
| | | *Adjusted R-squared* | | 0.2953 |

Table 4: ANOVA Results for gpt-4o-mini-2024-07-18

| Factor | Sum of Squares | Degrees of Freedom | F-Statistic | p-value |
|---|---|---|---|---|
| domain | 1.2389 | 11.0000 | 0.5004 | 0.6063 |
| perspective | 0.5032 | 5.0000 | 0.4472 | 0.5037 |
| 0-CoT | 30.9486 | 1.0000 | 137.5118 | 0.0000 |
| CAR | 156.8135 | 1.0000 | 696.7577 | 0.0000 |
| element:type | 224.0990 | 1995.0000 | 0.4991 | 0.6071 |
| Residual | 18025.1941 | 80090.0000 | | |
| | | *R-squared* | | 0.4436 |
| | | *Adjusted R-squared* | | 0.4430 |

Table 5: ANOVA Results for claude-3-5-sonnet-20240620

| Factor | Sum of Squares | Degrees of Freedom | F-Statistic | p-value |
|---|---|---|---|---|
| domain | 1.4703 | 11.0000 | 0.3814 | 0.5369 |
| perspective | 0.6683 | 5.0000 | 0.3814 | 0.5369 |
| 0-Cot | 0.2629 | 1.0000 | 0.7502 | 0.3864 |
| CAR | 0.4909 | 1.0000 | 1.4007 | 0.2366 |
| element:type | 266.6568 | 1995.0000 | 0.3814 | 0.5369 |
| Residual | 38842.9909 | 110826.0000 | | |
| | | *R-squared* | | 0.2336 |
| | | *Adjusted R-squared* | | 0.2330 |

Table 6: ANOVA Results for claude-3-haiku-20240307

# H MODELS

| Model Name | Model Card | Chat/ Instruction Tuned |
|---|---|---|
| **Closed-Source** | | |
| *OpenAI* | | |
| gpt-4o | | ✓ |
| gpt-4o mini | | ✓ |
| *Anthropic* | | |
| claude-3-5-sonnet | | ✓ |
| claude-3-haiku | | ✓ |
| *Huggy Llama* | | |
| llama-7b | huggyllama/llama-7b | ✗ |
| llama-13b | huggyllama/llama-13b | ✗ |
| llama-30b | huggyllama/llama-30b | ✗ |
| llama-65b | huggyllama/llama-65b | ✗ |
| *Meta Llama* | | |
| Llama-2-7b-hf | meta-llama/Llama-2-7b-hf | ✗ |
| Llama-2-13b-hf | meta-llama/Llama-2-13b-hf | ✗ |
| Llama-2-7b-chat-hf | meta-llama/Llama-2-7b-chat-hf | ✓ |
| Llama-2-13b-chat-hf | meta-llama/Llama-2-13b-chat-hf | ✓ |
| Llama-3-8B | meta-llama/Meta-Llama-3-8B | ✗ |
| Llama-3-8B-Instruct | meta-llama/Meta-Llama-3-8B-Instruct | ✓ |
| Llama-3.1-8B | meta-llama/Meta-Llama-3.1-8B | ✗ |
| Llama-3.1-70B | meta-llama/Meta-Llama-3.1-70B | ✗ |
| Llama-3.1-70B-Instruct | meta-llama/Meta-Llama-3.1-70B-Instruct | ✓ |

| Model Name | Model Card | Chat/Instruction Tuned |
|---|---|---|
| *Mistral* | | |
| Mistral-7B-v0.1 | mistralai/Mistral-7B-v0.1 | ✗ |
| Mathstral-7B-v0.1 | mistralai/Mathstral-7B-v0.1 | ✗ |
| Mistral-7B-v0.3 | mistralai/Mistral-7B-v0.3 | ✗ |
| Mistral-7B-Instruct-v0.3 | mistralai/Mistral-7B-Instruct-v0.3 | ✓ |
| Mistral-Nemo-Base-2407 (12.2B) | mistralai/Mistral-Nemo-Base-2407 | ✗ |
| Mistral-Nemo-Instruct-2407 (12.2B) | mistralai/Mistral-Nemo-Instruct-2407 | ✓ |
| *TIIUAE* | | |
| falcon-7B | tiiuae/falcon-7b | ✗ |
| falcon-11B | tiiuae/falcon-11B | ✗ |
| *AI21* | | |
| Jamba-v0.1 | ai21labs/Jamba-v0.1 | ✗ |
| AI21-Jamba-1.5-Mini | ai21labs/AI21-Jamba-1.5-Mini | ✗ |

Table 7: Overview of the open- and closed-source LLMs we evaluated. The table includes their names, their model card links, and whether they have been chat or instruction tuned. Models are grouped by family and sorted by parameter size, with non-chat-tuned models listed first within each group.

# I EXTRA RESULTS

## I.1 PERFORMANCE ON ELEMENTS GENERATED BY CLAUDE 3.5 SONNET

To assess whether the performance on our dataset was influenced by the choice of the generation LLM, we re-generated three elements from scratch using claude-3-5-sonnet. We selected Find Equilibrium Price because it exhibited the largest performance gap between gpt-4o and claude-3-5-sonnet, Diminishing Marginal Products was chosen as a random element with slight performance variation across the models, and Price Elasticity of Demand served as a control where no significant differences were expected.

Figure 8 shows the exact-match performance of both models on these three elements. We found no significant differences in performance between any of the models.

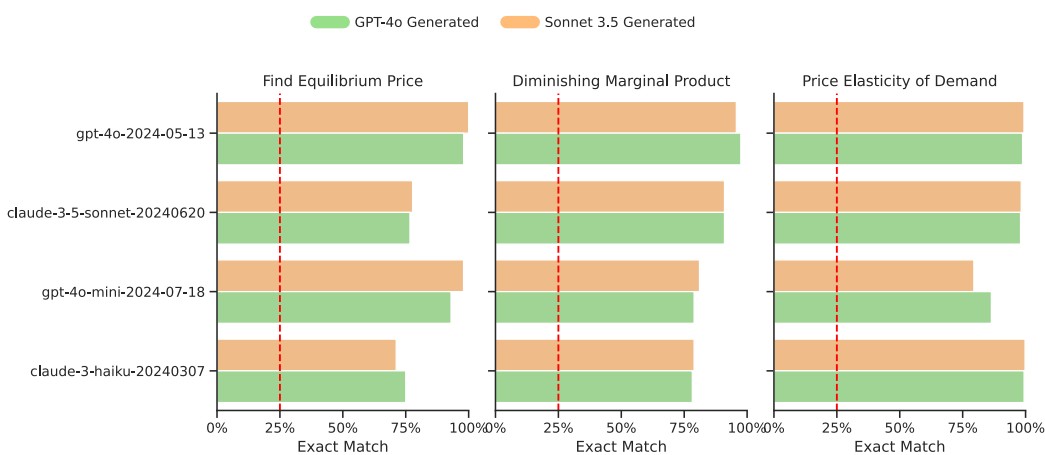

Figure 8: Exact-match performance comparison between closed-source models on three that were generated by gpt-4o and claude-3-5-sonnet. The elements were selected based on the observed performance differences across models, with Price Elasticity of Demand serving as a control. The figure shows no significant differences in performance between the models on these elements. Note that the red dotted line signifies random guessing performance.

## I.2 DEADWEIGHT LOSS

To conduct our error analysis, we ran all closed-source models on the free-text QA adaptation of the deadweight loss task. We began by inspecting a range of model outputs to identify distinct classes of errors that were common across responses. Once these error categories were established, we computed the answers corresponding to these errors and rescored the models based on whether their outputs were within 98% of either the correct answer or any of the answers derived from specific error assumptions. We also ensured that when there was any overlap in incorrect responses that we chose the closest one to the model's response. This approach allowed us to capture not only the frequency of correct outputs but also the systematic nature of the models' reasoning flaws. Below, we provide a detailed breakdown of the primary error types:

- Incorrect Base for Deadweight Loss Type 1: This error incorrectly substitutes $P_e - P_m$ (the difference between the competitive equilibrium price and the monopolist's price) in place of the correct term $P_m - MC(Q_m)$ (the difference between the monopolist's price and the marginal cost at the monopolist's quantity).
- Incorrect Base for Deadweight Loss Type 2: This error calculates the deadweight loss using the difference between the monopoly price and the competitive equilibrium price as the base of the triangle.
- Incorrect Base and Height Type 1: This error replaces the base of the DWL triangle $(Q_e - Q_m)$ with a miscalculated value for the equilibrium quantity and replaces the base with the Type 1 variant.

- Incorrect Base and Height Type 2: This error assumes that $P_e = MC(Q_m)$, leading to an incorrect height calculation where the DWL triangle's height becomes $P_m - P_e$.
- Treating Marginal Cost as Constant: Instead of recognizing marginal cost as a function derived from the supply curve ($MC(Q) = a_{\text{supply}} \times Q + b_{\text{supply}}$), several models treated marginal cost as a constant, often equal to the slope of the supply curve. This assumption led to errors in determining the monopolist's quantity and price, further propagating inaccuracies in the deadweight loss calculation.
- Combined Errors: A subset of models combined the two errors above, simultaneously using an incorrect formula for deadweight loss and assuming a constant marginal cost. This compounded error significantly reduced the likelihood of producing a correct answer and highlighted the systematic nature of the misunderstanding.

### I.3 PERFORMANCE GAPS BETWEEN FREE TEXT QA AND HIDDEN MCQA

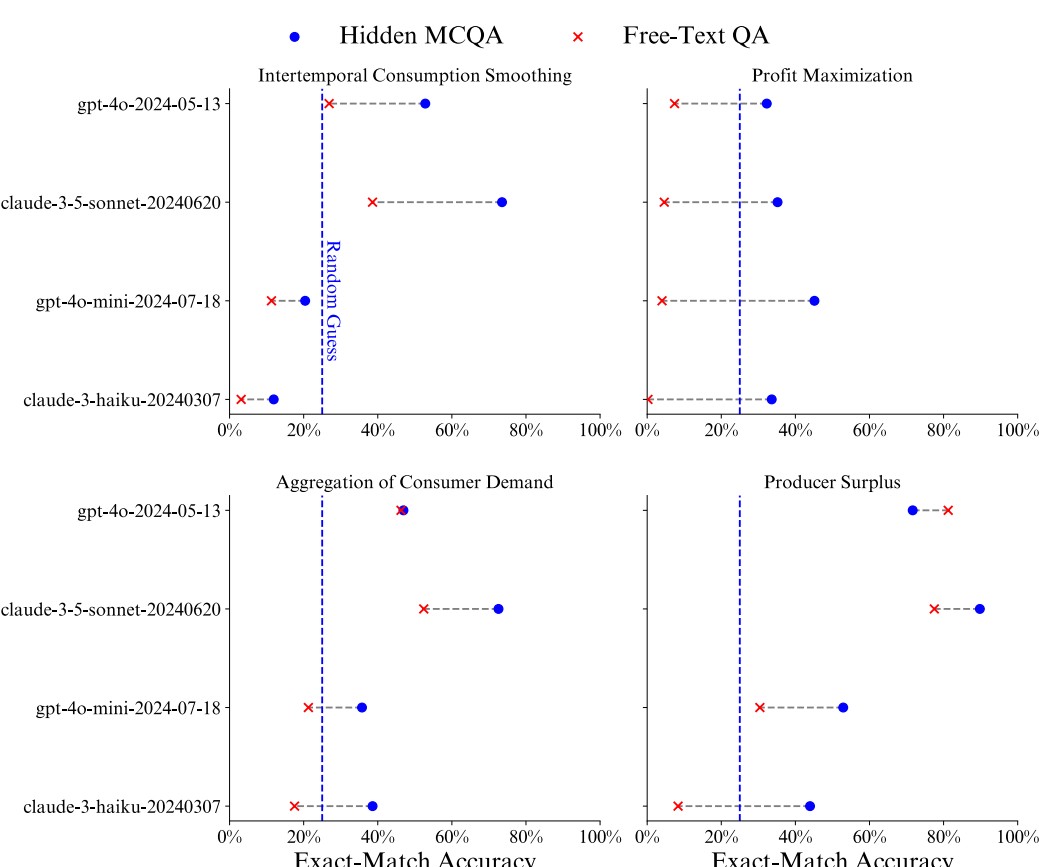

Figure 9: Comparison of exact-match accuracy for four elements (Intertemporal Consumption Smoothing, Profit Maximization, Aggregation of Consumer Demand, and Producer Surplus) across all closed-source models. The plot illustrates the difference in performance under the hidden and free-text adaptations, highlighting the impact of multiple-choice options on reasoning accuracy. Plotted in blue is the accuracy random guessing would have in the hidden adaptation, note that this line is not relevant for the free-text QA adaptation.

While most cases show a negative gap between hidden and free-text QA performance, there are notable exceptions. Figure 9 shows that in the Producer Surplus element, gpt-4o performed better in the free-text QA adaptation compared to the hidden adaptation. One might expect that the explanation for this positive gap is that the multiple-choice answers in the hidden adaptation were very similar, which may have caused confusion for the LLM. However, our analysis shows that when the free-text QA was scored correctly, gpt-4o selected an incorrect answer in the hidden adaptation with

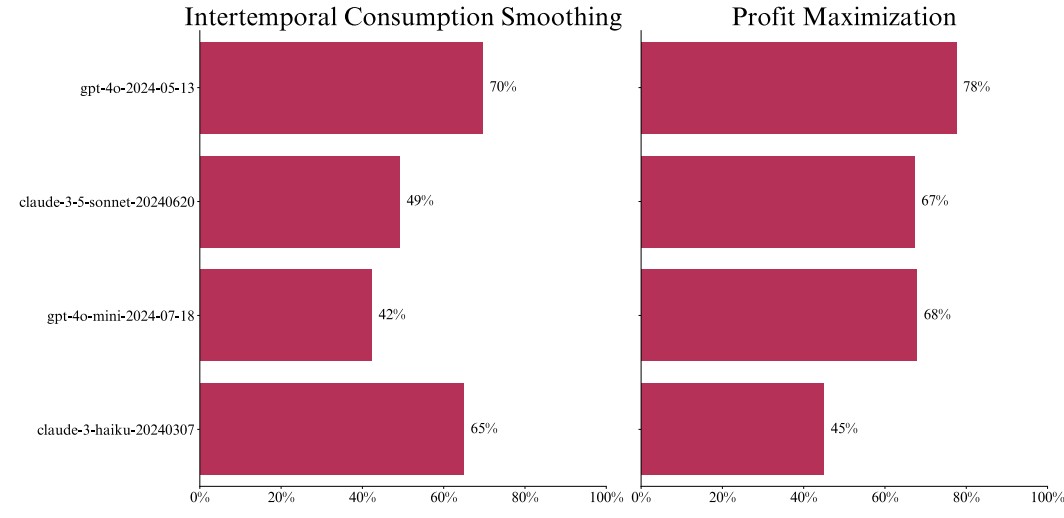

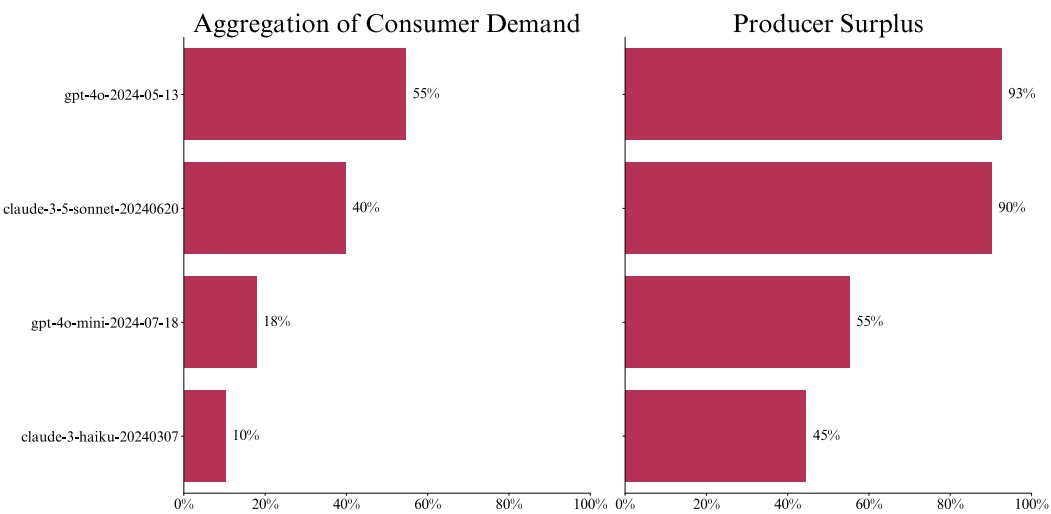

Figure 10: This figure depicts the percentage of time models were incorrect in the free-text adaptation but correct on the hidden adaptation due to choosing the closest answer. The plot compares the performance of four elements—Intertemporal Consumption Smoothing, Profit Maximization, Aggregation of Consumer Demand, and Producer Surplus—across all closed-source models.

at least a 10% difference from the correct answer 78% of the time. This suggests that while the LLM could reason effectively about the problem, it struggled to correctly match its reasoning to the multiple-choice options provided.

The plot also suggests that the performance gap in the Profit Maximization element is primarily due to the benefit random guessing has on accuracy in the hidden adaptation compared to free-text QA. Furthermore, in the Aggregation of Consumer Demand element, the inclusion of options after the reasoning step offered limited benefit, highlighting that the true advantage lies in including these options during the reasoning process.

These observations highlight an important nuance: although multiple-choice formats generally offer helpful structure for models, they may also hinder performance in certain scenarios.

# J  SECTION 3 IMAGES

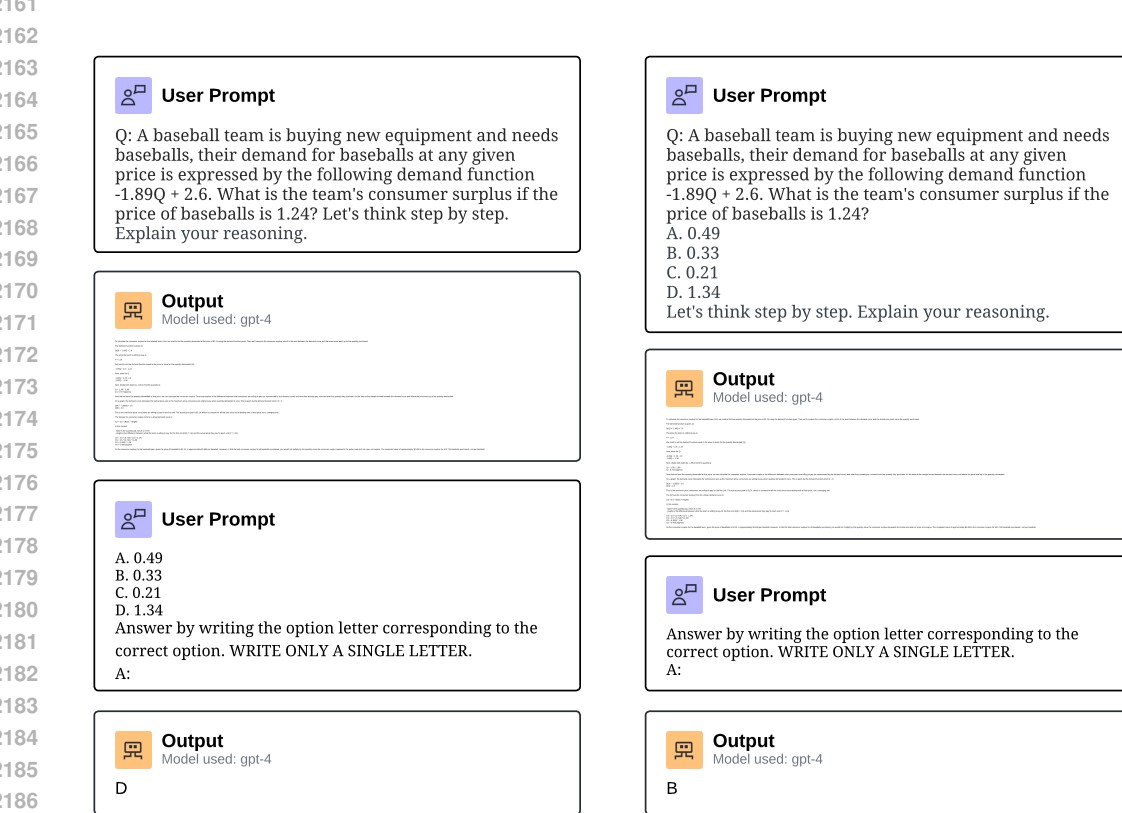

Figure 11: (Left) The hidden approach to 0-CoT: the model is given only the question and asked to explain its reasoning before being provided with options. (Right) The shown approach to 0-CoT: the model is presented with both the question and options before explaining its reasoning.

# K  WEB APPLICATION

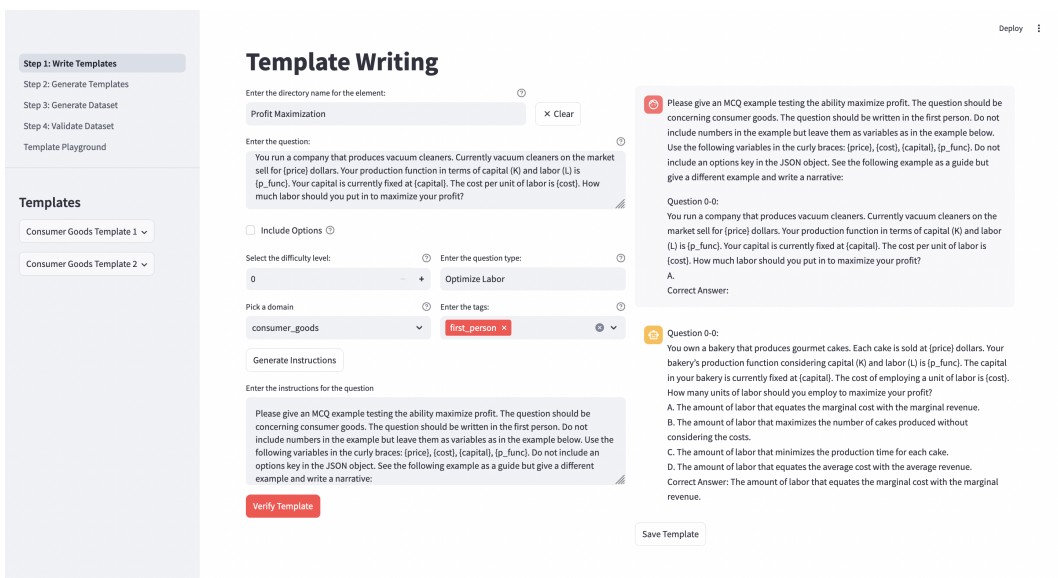

Figure 12: The web app user interface for template writing. This page includes fields for type, domain, grade level and tags (including perspectives). The right shows an example of template verification which uses a LLM to generate another template using the example seed.

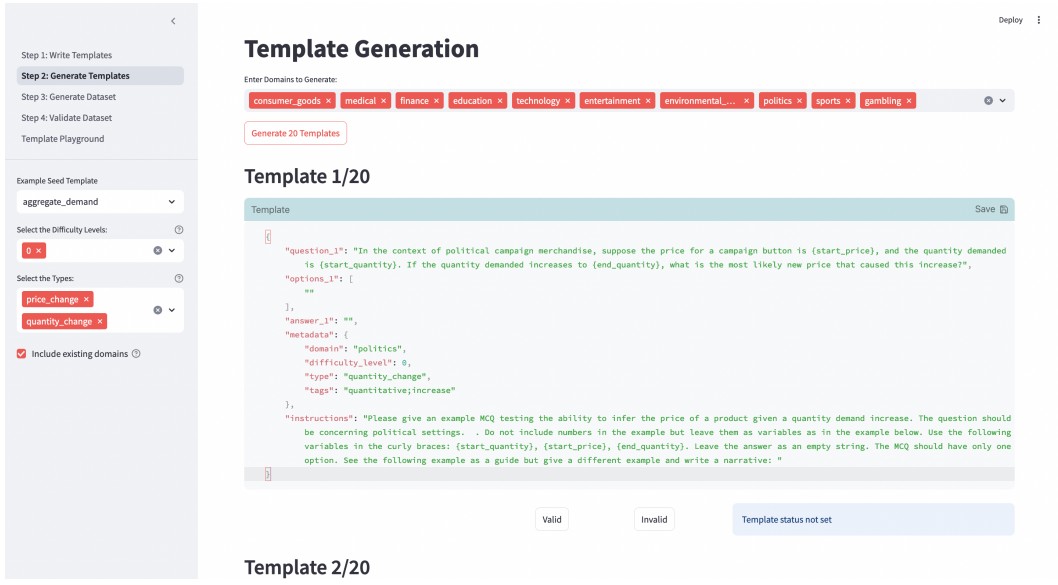

Figure 13: The web app user interface for template generation. This page allows for a selection of domains, and types for which templates will be generated using the available example seeds. Templates can then be verified and saved by the user.