# OpenReview forum: "STEER-ME: Assessing the Microeconomic Reasoning of Large Language Models"
_ICLR.cc/2025/Conference — Submitted to ICLR 2025_

### Official Review · Reviewer_VLWY · 2024-10-28

**Soundness:** 1
**Presentation:** 2
**Contribution:** 1
**Rating:** 5
**Confidence:** 4

**Summary:**

The paper introduces a novel benchmark, STEER-ME, to assess LLM's microeconomic reasoning in non-strategic settings. Using an LLM-assisted data generation method, Auto-STEER, the benchmark includes 57 distinct microeconomic elements across multiple domains and perspectives, minimizing data contamination. The authors evaluated 15 LLMs on these benchmarks, highlighting strengths and limitations in handling non-strategic economic decision-making.

contributions:
1. Novel benchmark development: The STEER-ME benchmark assesses LLM performance in non-strategic microeconomics, a domain not extensively covered by previous benchmarks that focused mainly on game theory or strategic decision-making.

2. Comprehensive taxonomy: The paper categorizes microeconomic reasoning into 57 elements, each tested across diverse contexts, such as finance and public policy, to evaluate broad economic reasoning capabilities.

**Strengths:**

1. proposed a benchmark on economics reasoning, which is relatively novel compared to classical benchmarks.

2. evaluated quite a few LLMs on the benchmark and have achieved some results.

**Weaknesses:**

Section 2: There are numerous economic concepts that may be challenging for machine learning researchers to follow, particularly the rationale behind constructing the taxonomy shown in Figure 1.

Section 3.1: Details about the dataset are lacking. Key statistics, such as the distribution of questions and question lengths, are not provided. Additionally, there is no discussion on how the dataset’s quality is ensured.

Section 5: The analysis of experimental results is limited. There is no case study or error analysis presented to offer deeper insights into model performance.

Code / Data:  i have not seen it uploaded.

**Questions:**

See weakness. In addition, i have questions below

1. Why is it necessary to develop these benchmarks for LLMs? Existing finance-related benchmarks, such as FinQA, FinEval, FinBench, FinanceBench, and FAMMA, already appear to cover similar topics to those in STEER. Additionally, general reasoning datasets in common scenarios may address some overlapping subjects.

2. What specific insights does this benchmark provide for LLM researchers?



--------------------------------------------------
I raise to 5 for the clarification about the motivation and dataset.

---

> ### Author Response · Authors · 2024-11-26
>
> ## Weaknesses
> 1. We acknowledge that the summary level in the main text is brief. To address this, we provide a more detailed rationale in Appendix A, where we explain how we selected and grouped the elements included in our benchmark. While this discussion offers additional depth, it is not intended to exhaustively capture the full scope of the economic literature. For readers interested in further exploration, we include references to foundational texts that provide a comprehensive treatment of these concepts and classifications.
>
> 2. We appreciate the reviewer's comment and have addressed these concerns in the resubmitted paper. To provide greater transparency, we have included additional summary statistics about the dataset in Section 3.1. These statistics detail the total number of elements, the average and median character counts per question, and the distribution of questions across domains and types, offering a clearer picture of the dataset’s composition and scope.
>
>     To ensure the quality of the dataset, we implemented a two-step verification process. First, all style-transferred templates were manually reviewed by a human to confirm their economic and semantic correctness. Second, we spot-checked a sample of the generated templates to assess their validity. While we did not have the resources to verify every generated question, the high success rate observed during spot-checking gives us confidence in the overall quality of the dataset. We have clarified these quality assurance measures in the auto-STEER section to highlight their role in maintaining the reliability of the benchmark.
>
> 3. We appreciate the reviewer’s feedback regarding the need for a deeper analysis of experimental results. In response, we have expanded Section 5 in the revised version to include detailed case studies and error analyses, highlighting both the strengths and limitations of LLMs on our benchmark.
>
>     One key addition is our analysis of free-text QA. This adaptation revealed notable challenges: while models often approximate correct answers in multiple-choice settings by recognizing and selecting the most plausible option, their performance in free-text QA declines. This is particularly evident in tasks requiring precise economic reasoning, where models frequently struggle to articulate complete and accurate responses.
>
>     We also observed that including multiple-choice options can positively condition models toward engaging with the question more effectively, as these options serve as cues that guide the reasoning process. In contrast, the open-ended format removes these cues, exposing areas where models lack depth in reasoning or misinterpret economic concepts.
>
> ## Questions
> 1. We appreciate the opportunity to clarify the motivation for developing STEER-ME and how it differs from existing benchmarks.
>
>     First, many existing finance-related benchmarks, such as FinanceBench, FiQA, and FinEval, focus primarily on classification-oriented tasks, such as sentiment analysis, named entity recognition, document classification, or knowledge-based question answering. While these benchmarks are valuable for specific NLP applications, they do not evaluate a model’s reasoning capabilities in economic contexts. STEER-ME, by contrast, is designed to assess reasoning skills, requiring models to engage with complex economic problems rather than simply categorizing or retrieving information.
>
>     Finally, while general reasoning datasets may overlap with some of the reasoning types STEER-ME investigates, they lack the depth and specificity required for evaluating economic reasoning. These datasets often focus on general problem-solving or abstract reasoning but do not delve into domain-specific economic concepts. STEER-ME goes beyond testing definitional recall to evaluate whether models can apply economic principles to answer complex, context-dependent questions. This includes rephrased and domain-specific questions that require adaptability and reasoning in specialized economic contexts, making STEER-ME distinct from broader reasoning benchmarks.
>
> 2. Please refer to the overall meta-comment.

---

### Official Review · Reviewer_4BWC · 2024-10-30

**Soundness:** 3
**Presentation:** 3
**Contribution:** 3
**Rating:** 6
**Confidence:** 3

**Summary:**

This study fills a gap in the assessment of existing Large Language Models (LLMs) for non-strategic microeconomic reasoning tasks. To achieve this goal, the authors propose a new STEER-ME benchmark that categorizes microeconomic reasoning into 57 elements, generates a large number of multiple-choice questions and dynamically generates test data using the LLM-assisted data generation protocol (auto-STEER) to reduce the risk of data contamination. The experimental results show that the performance of different LLMs in the microeconomic reasoning task varies significantly, with even the larger models falling short; the only model that consistently performs well is o1-preview.

**Strengths:**

1. This work identifies a gap in the evaluation of current LLMs for non-strategic microeconomic reasoning tasks with high practical needs and research value.

2. A structured STEER-ME benchmark and a dynamic data generation protocol auto-STEER are proposed to ensure test diversity and data cleanliness.

3. The experiments cover a wide range of models and adaptation strategies, revealing differences in the performance of different LLMs in non-strategic microeconomic reasoning.

**Weaknesses:**

1. In the AUTO-STEER section, the authors may need to clarify which type of LLM was used for data generation. Additionally, it is important to discuss the impacts of the benchmark results by different LLMs for data generation.

2. The authors note that LLMs struggle with basic mathematical problems, like calculating the deadweight loss of a monopoly[Line 445-450], and suggests that this may be attributed to the use of incorrect formulas. To strengthen this claim, the authors should present specific evidence, such as analyses of error cases.

3. The authors need to provide more information on which open-source LLMs were tested (they did not list the LLMs), particularly whether they included models from the fields of mathematics and economics. It would also be important to discuss any differences in results between these specialized LLMs and general LLMs.

**Questions:**

1. It would be better to explore the effects of LLMs under in-context learning conditions.

---

> ### Author Response · Authors · 2024-11-26
>
> ## Weaknesses
>
> 1. Thanks for catching this! We used gpt-4o-2024-05-13 and have updated that in the paper. We also have added a case study on 3 elements chosen for their variance between claude and gpt models that were fully generated by claude-3-5-sonnet in Appendix I.2. We find that the claude-3-5-sonnet generated elements do not induce a performance difference downstream.
>
> 2. This is a great point. We add a more in-depth discussion into the error cases for deadweight loss in the results section and the appendix. At a high level, we computed answers that would have arisen given specific incorrect formulas and identify when the model's answer is closest to the correct answer or the incorrect answer (with some minimum tolerance). We found that in the vast majority of cases (\~82\% of the time) claude-3-5-sonnet used four easily-identifiable incorrect formulas and in the majority of cases (\~54\%) gpt-4o used those same incorrect formulas.
>
> 3. In Table 2 in the appendix we listed the 15 LLMs we evaluated. We have updated that number since the original submission to be 27 total LLMs with 22 being open-source LLMs. This did include some LLMs tuned specifically for mathematics (e.g., Mathstral) and we did not find much difference in performance on these models.
>
> ## Questions
> 1. We appreciate the reviewer's suggestion and agree that exploring the effects of in-context learning is essential to understanding LLM performance. In fact, our benchmark explicitly incorporates in-context learning scenarios by presenting LLMs with prompts that frame economic reasoning tasks within a broader context. These prompts include relevant background information to simulate realistic economic problem-solving conditions. Additionally, we analyze performance across different domains to identify whether and how contextual framing influences model behavior.

---

> > ### Comment · Reviewer_4BWC · 2024-11-26
> >
> > Thank you for your reply. You've answered most of my questions.

---

### Official Review · Reviewer_iHDk · 2024-10-31

**Soundness:** 3
**Presentation:** 3
**Contribution:** 3
**Rating:** 6
**Confidence:** 3

**Summary:**

The paper presents a new benchmark specifically designed to evaluate large language models (LLMs) in non-strategic microeconomic tasks, addressing a gap in existing benchmarks that focus more on strategic decision-making. The authors create a taxonomy of 57 microeconomic reasoning elements across multiple domains, perspectives, and types, and introduce a data generation protocol called auto-STEER to produce fresh, contamination-resistant questions for each element. By evaluating 15 LLMs using this benchmark, the study reveals strengths and weaknesses in the models' ability to handle microeconomic concepts such as optimization and marginal analysis.

**Strengths:**

1. The creation of a taxonomy with 57 elements specifically for non-strategic economics provides a strong theoretical basis for the benchmark. This comprehensive taxonomy ensures that the benchmark is grounded in a thorough understanding of microeconomic reasoning, enabling a detailed and structured evaluation of LLMs in this domain.
2. The benchmark is solid, due to its broad variation in testing angles and use of multiple testing metrics.

**Weaknesses:**

1. The paper fails to discuss the correlation between the results of the proposed benchmark and those of related benchmarks, highlighting the significance of this research.
2. There are still some issues that need to be addressed in the writing of the paper, such as:
- The paper lacks hyperlinks for tables, figures, and sections, making it difficult for readers to locate referenced content.
- The title of Figure 2 contains misleading positional descriptors for the two images, referring to them as ‘on top’ and ‘on the right.’

**Questions:**

1. The analysis in the results section appears incomplete. Why is there only a separate section on "Domain Robustness" without any mention of "type robustness" or "perspective robustness"?
2. Although the paper generates a large number of questions for testing purposes, is this amount really necessary? For instance, the paper creates 100 templates by rephrasing, only modifying exact words or objects in the questions. Will these modifications lead to significant differences in results?

---

> ### Author Response · Authors · 2024-11-26
>
> ## Weaknesses
> 1. Thank you for your comment. While direct comparisons of results are challenging due to the distinct focus of STEER-ME, we agree that providing a discussion of how it relates to other benchmarks would strengthen the paper. In the revised version, we have better highlighted how STEER-ME fits within the broader ecosystem of benchmarks. This discussion emphasizes STEER-ME's unique contributions to evaluating reasoning in non-strategic microeconomics and clarifies its role and significance in advancing research in this area.
>
> 2. Thanks! We have made these corrections.
>
> ## Questions
>
> 1. Thank you for pointing out this omission. In the resubmitted paper, we have conducted an analysis on type robustness. We offer a brief summary of the results. We investigated model performance across different functional families in the questions. We observed that performance on Cobb-Douglas functions, which are widely used in economics and involve real-valued exponents, was worse compared to functions with integer-valued exponents. However, despite challenges with real-valued exponents, models performed equally well on both Cobb-Douglas and linear functions for the output elasticity and returns to scale elements. We hypothesize that this is because there are simple solutions to Cobb-Douglas production functions in these elements: the output elasticity is simply the exponent, and the returns to scale is the sum of the exponents.
>
> 3. Our results indicate that varying the text of questions, even when conditioning on fixed attributes such as element, type, domain, and perspective, leads to significant variation in LLM performance. This suggests that rephrased templates are not trivial modifications and provide meaningful insights into the model's ability to generalize across semantically equivalent formulations. To quantify this, we conducted experiments (detailed in Appendix G) and observed that models exhibit variability in performance across these rephrased templates, underscoring the importance of including such variability in evaluations.
>
>     One of the main motivations for generating a large number of templates is to capture this variability, as studies (e.g., Mirzadeh et al., 2024) have shown that rephrasings can significantly affect model performance. Additionally, when sampling answers from models at temperature 0 (standard for analytical tasks), responses are deterministic. To achieve statistical power, it is essential to test the model with numerous rephrased questions. This is akin to asking 100 humans the same question.

---

> > ### Comment · Reviewer_iHDk · 2024-11-26
> >
> > Thank you for your feedback. Most of my concerns have been resolved, and thus I will maintain positive score.

---

### Official Review · Reviewer_8XWi · 2024-11-04

**Soundness:** 3
**Presentation:** 3
**Contribution:** 2
**Rating:** 5
**Confidence:** 3

**Summary:**

This paper presents a new benchmark specifically tailored to evaluate LLMs' performance in non-strategic microeconomic reasoning. This benchmark, STEER-ME, addresses limitations in existing economic benchmarks by categorizing microeconomic reasoning tasks into 57 distinct elements, covering various domains and perspectives. The authors also propose an LLM-assisted data generation protocol that dynamically generates questions to prevent data contamination. The study evaluates 15 LLMs across these elements to identify both strengths and limitations in their reasoning capabilities, providing insights into current LLM performance on fundamental economic tasks.

**Strengths:**

1. The paper offers a highly comprehensive benchmark, breaking down microeconomic reasoning into 57 distinct elements that span a wide range of domains and perspectives.
2. Significant resources and computational effort were invested in the study, with $5,896.33 spent on API requests to OpenAI and Anthropic, and 6.81 GPU years of compute used to evaluate open-source models.
3. All model outputs are made publicly available, promoting reproducibility and enabling further research and contributions to evaluation practices.
4. The study employs a wide range of evaluation metrics, including calibration-related metrics like Expected Calibration Error (ECE).
5. The paper highlights the critical issue of data contamination in LLM benchmarks

**Weaknesses:**

1. The primary contribution—expanding benchmarks to cover non-strategic microeconomic settings—leans more toward an economic contribution than a machine learning one, which may affect the paper's resonance with ICLR’s core audience.
2. The paper asserts that auto-STEER addresses data contamination but lacks empirical evidence or a detailed explanation of how it mitigates this problem effectively.
3. The benchmark is limited to multiple-choice questions, whereas real-world financial assistant LLMs are often required to handle open-ended generation tasks relevant to economic analysis. I would recommend expanding more experiments with more types of datasets, such as the free-text generation QA datasets.
4. The extensive number of elements and perspectives, while thorough, could complicate the task of providing a clear, overall model recommendation.
5. A substantial portion of the paper focuses on introducing economic rationality elements, which might not align well with the technical expectations of ICLR’s machine-learning audience. But overall, the idea is pretty interesting.

I am willing to raise my score if the paper could include more analysis (e.g., more evaluations on different types of datasets) on the perspectives of LLM-wise or machine learning-wise.

**Questions:**

Please refer to the weakness.

---

> ### Author Response · Authors · 2024-11-26
>
> 1. We appreciate the reviewer’s perspective and would like to clarify that our work, while grounded in economic concepts, is fundamentally aimed at advancing machine learning applications. By evaluating LLMs in non-strategic microeconomic contexts, we tackle unique challenges in structured decision-making under constraints. These tasks provide a rich test bed for assessing LLMs' reasoning capabilities, particularly their ability to integrate mathematical reasoning, domain-specific knowledge, and contextual understanding. This focus aligns with the broader machine learning goal of understanding and improving general reasoning abilities in AI systems.
>
>    Economic reasoning, like logical reasoning, is implicit in many decision-making problems for agents across diverse domains. If an LLM is deployed as an agent reasoning about markets or changes in prices, the capabilities evaluated by this benchmark would directly inform its effectiveness. Furthermore, our benchmark provides a framework for examining LLMs’ reasoning in a domain that combines abstraction with real-world complexity, making it broadly relevant to machine learning research.
>
>     Our methodology aligns with the trend in the ML community of using domain-specific benchmarks to assess reasoning capabilities. Similar to SWEBench (software engineering tasks, ICLR 2023), BEND (biology-focused reasoning, ICLR 2024), and ISARSTEP (high-level mathematics, ICLR 2021), STEER-ME leverages a complex domain to rigorously evaluate model generalization and reasoning limitations. These works demonstrate that domain-specific benchmarks can yield insights into model capabilities that extend beyond the evaluated domain, and we view STEER-ME as a meaningful contribution within this tradition.
>
> 2. Thank you for raising this concern. While a direct investigation of auto-STEER's ability to mitigate data contamination is beyond the scope of this paper, we appreciate the need to provide context for its effectiveness. In Section 3.2, we describe how auto-STEER generates diverse questions by systematically varying templates, domains, and perspectives. To address your comment, we have added a discussion in Appendix B that draws on related work to support our design. For instance, Mirzadeh et al. (2024) show that modifying question templates by swapping names and numbers—an approach similar to auto-STEER—significantly reduces performance on the GSM8K benchmark, highlighting its susceptibility to contamination. This evidence reinforces the potential of auto-STEER to address data contamination effectively, and we hope this additional context clarifies its motivation and design.
>
> 3. We appreciate the reviewer's suggestion to incorporate open-ended generation tasks as part of our evaluation framework. In the resubmitted paper, we have addressed this by performing a case study on the performance of closed-source models in a free-text generation QA adaptation, applied to a subset of elements.
>
>     To examine the effect of multiple-choice formats, we selected 4 elements (one from each non-Foundations setting) with the largest gap in accuracy between the hidden and shown adaptations to evaluate in free-text QA. Our analysis revealed that performance is consistently worse under the free-text adaptation compared to the hidden adaptation. This performance gap appears to stem from the models' tendency to approximate correct answers: in the multiple-choice setting, they often recognize and select the closest option, whereas in free-text generation, this heuristic is less effective.
>
> 4. Our goal is not to recommend a single best model but to offer a variety of perspectives that allow users to select a model that best suits their particular use case. The extensive elements and cross-cutting analyses are intended to provide insights into model performance across different economic reasoning tasks. That being said, we highlight that o1-preview is the best performer in all of the elements we tested, but ultimately, the goal is to offer a nuanced understanding of strengths and weaknesses rather than a one-size-fits-all recommendation.
>
> 5. We appreciate this suggestion and have made revisions to address this feedback. In the resubmitted paper, we include additional analysis focused on free-text QA. If this additional analysis does not fully align with the reviewer's expectations, we would be happy to discuss any further evaluations or clarify specific aspects of interest.

---

> > ### Comment · Reviewer_8XWi · 2024-12-03
> > **Official comments of Reviewer 8XWi**
> >
> > Thanks for you detailed reponses. The authors have addressed partial of my concerns. However, the case studies are not sufficient to prove the effectiveness of free-text generation QA dataset. It is highly recommended to conduct an quantitive evaluation instead of purely showing qualitative results. One evaluation results on single dataset is not sufficent to convince the audience of ICLR, including myself. I will thus keep my score.
> >
> > I encourage the authors to further refine their work. This is a well-written paper on a very interesting topic, and I believe it will become a very good and solid work with appropriate revisions.

---

> > > ### Author Response · Authors · 2024-12-03
> > >
> > > Thank you for your follow-up comments and for recognizing the potential of our work. We understand your concern regarding the need for more quantitative evaluation of the free-text QA adaptation. To clarify, we have collected full model outputs across all elements and models for all three datasets: MCQA shown, MCQA hidden, and free-text QA. However, while we have extracted and reported scores for the MCQA datasets, obtaining the answers from the free-text QA responses is costly because we have to query another LLM to extract the answers from the outputs. This was the primary reason we focused on the MCQA datasets for comprehensive scoring in this submission.
> > >
> > > Despite our limited resources, we analyzed free-text QA responses to better understand how multiple-choice options influence performance. Specifically, we examined cases where the performance gap between MCQA shown and MCQA hidden indicated reliance on multiple-choice options. Our analysis revealed two key insights. First, LLMs often gained performance in MCQA by selecting the closest option, even when their underlying free-text answer would not meet the tolerance for correctness. Second, we observed instances where the LLM would map their free-text answer to the incorrect multiple choice option even when their answer was correct as per free-text QA.
> > >
> > > We hope this clarifies our approach and the insights gained from the free-text QA analysis, despite the challenges in fully quantifying its results. Thank you again for your thoughtful feedback and constructive suggestions.

---

### Author Response · Authors · 2024-11-26

We would like to thank all reviewers for their thoughtful comments and for highlighting areas where our work could be further improved. We recognize that some aspects of our benchmark's insights may not have been fully clear, and we appreciate the opportunity to elaborate. Here, we outline some of the key insights our initially submitted work offered:

1. Reasoning over Options: One particularly interesting finding is that LLMs demonstrate an ability to recognize when a question can be resolved using the provided options. This allows them to "short-circuit" difficult computations by leveraging the structure of multiple-choice questions, a behavior that suggests nuanced reasoning strategies.
2. Challenges with Real-Valued Exponents: Real-valued exponent computations, which are prevalent in economic reasoning (e.g., interest rates, discounting, elasticities), present a significant challenge for LLMs. This reveals a gap in the models' ability to handle key mathematical operations that are critical for solving common economics problems.
3. Domain-Specific Conditioning: We also observe that certain domains influence LLM performance. For example, when questions are framed in more realistic, real-life scenarios, models often fail to recognize what is being asked. They tend to equivocate or misinterpret the problem, indicating potential biases introduced by the contextual framing of the task.

---

### Author Response · Authors · 2024-12-03
**Highlights from the Resubmitted Paper**

# Additions to the Evaluation:

1. We have tested 12 additional LLMs as listed in the table in the appendix.
2. We have created an additional adaptation, free-text QA, and have evaluated all models across all elements on this adaptation.

# Additions to the Results:


1. Conducted a quantitative analysis on the impact of multiple-choice options on chain-of-thoughts showing that in nearly all cases for all of the closed-source models that offering options increase a model's ability to answer correctly.
2. Conducted case studies diving deeper into the impact of MCQA on performance

    a. Showed that in elements that dealt with aggregation, the inclusion of options aided the LLMs' abilities to condition on the right reasoning path.

    b. Offered a number of new insights through the free-text QA adaptation on a subset of elements that had high performance gaps between MCQA hidden and shown.
3. Added a deeper case/error analysis of the mathematical errors on the deadweight loss element using the free-text QA adaptation as the diagnostic tool. We then characterized and detailed the specific error cases.
4. Added a type robustness section diving deeper into the issues LLMs had with different commonly used functional families.
5. Showed that the choice of generation LLM had negligible impact on downstream performance.

---

### Comment · Area_Chair_4NY1 · 2024-12-04
**End of reviewer-author discussion phase**

Dear reviewers,

As we near the conclusion of the reviewer-author discussion phase, I wanted to kindly follow up to see if you’ve had a chance to review the author responses on your comments. Could you confirm that you’ve read it and, if needed, update your review and scores accordingly?

Thank you for your time and effort!

Your AC

---

### Meta-Review · Area_Chair_4NY1 · 2024-12-22

**Metareview:**

Summary:
The paper introduces STEER-ME, a benchmark for evaluating LLMs' microeconomic reasoning capabilities in non-strategic settings. Key claims include:
- A taxonomy of 57 distinct microeconomic reasoning elements across multiple domains and perspectives
- Introduction of auto-STEER, an LLM-assisted data generation protocol to reduce data contamination risks
- Evaluation of 15 LLMs (later expanded to 27) showing varying performance levels in microeconomic reasoning tasks
- Finding that even larger models struggle with certain economic reasoning tasks, with o1-preview being the only consistently strong performer
- Discovery that LLMs particularly struggle with real-valued exponent computations and certain mathematical operations critical for economics

Strengths:
- Comprehensive taxonomy breaking down microeconomic reasoning into well-defined elements
- Significant computational resources invested ($5,896.33 on API requests, 6.81 GPU years)
- Public availability of model outputs promoting reproducibility
- Diverse evaluation metrics including calibration metrics
- Novel approach to generating fresh questions through auto-STEER protocol
- Broad coverage of domains and perspectives in economic reasoning

Weaknesses:
-  Limited quantitative evaluation of free-text QA adaptation, with analysis restricted to case studies
-  Insufficient empirical evidence demonstrating auto-STEER's effectiveness in mitigating data contamination
- Primary focus on multiple-choice questions may not fully reflect real-world economic reasoning tasks
- Complex taxonomy with numerous elements may complicate clear model recommendations
- Lack of comprehensive correlation analysis with related benchmarks
- Limited discussion of specialized economic LLMs' performance compared to general models

Reasons for Rejection:
-  Methodology Limitations: The heavy reliance on multiple-choice questions without sufficient quantitative evaluation of free-text generation limits the benchmark's practical applicability.
- Incomplete Validation: The paper lacks robust empirical evidence supporting auto-STEER's effectiveness in preventing data contamination.
- Scope vs. Depth Trade-off: While comprehensive in breadth, the large number of elements dilutes the depth of analysis for each component.
- Technical Innovation: The contribution leans more toward economic taxonomy than advancing machine learning methodology.

**Additional Comments On Reviewer Discussion:**

During the discussion phase, reviewers raised several key concerns about the paper's machine learning contributions, quantitative analysis depth, benchmark necessity, and mathematical error handling. The authors responded by expanding their evaluation to include 12 additional LLMs and introducing a free-text QA adaptation. They conducted deeper analyses of mathematical errors, particularly in deadweight loss calculations, and demonstrated that the choice of generation LLM had minimal impact on results. They also clarified how their benchmark differs from existing finance-related evaluations.
While these responses addressed some concerns, fundamental issues remained unresolved. The free-text QA evaluation stayed largely qualitative, and the authors didn't fully justify the benchmark's practical advantages over existing alternatives. Additionally, the limited technical innovation in machine learning methodology remained a concern. Though the authors made sincere efforts to improve the work, these persistent limitations support the rejection decision. The research direction shows promise, but the current submission would benefit from further development to address these core issues.

---

### Decision · Program_Chairs · 2025-01-22

Reject